# ⚇ DigiRL: Training In-The-Wild Device-Control Agents with Autonomous Reinforcement Learning

**Hao Bai**[1,2*]  **Yifei Zhou**[1*]  **Mert Cemri**[1]  **Jiayi Pan**[1]

**Alane Suhr**[1]  **Sergey Levine**[1]  **Aviral Kumar**[3,4]

[1]UC Berkeley  [2]UIUC  [3]CMU  [4]Google DeepMind

## Abstract

Training corpuses for vision language models (VLMs) typically lack sufficient amounts of decision-centric data. This renders off-the-shelf VLMs sub-optimal for decision-making tasks such as in-the-wild device control through graphical user interfaces (GUIs). While training with static demonstrations has shown some promise, we show that such methods fall short for controlling real GUIs due to their failure to deal with real world stochasticity and non-stationarity not captured in static observational data. This paper introduces a novel autonomous RL approach, called DigiRL, for training in-the-wild device control agents through fine-tuning a pre-trained VLM in two stages: offline RL to initialize the model, followed by offline-to-online RL. To do this, we build a scalable and parallelizable Android learning environment equipped with a VLM-based evaluator and develop a simple yet effective RL approach for learning in this domain. Our approach runs advantage-weighted RL with advantage estimators enhanced to account for stochasticity along with an automatic curriculum for deriving maximal learning signal. We demonstrate the effectiveness of DigiRL using the Android-in-the-Wild (AitW) dataset, where our 1.3B VLM trained with RL achieves a 49.5% absolute improvement – from 17.7 to 67.2% success rate – over supervised fine-tuning with static human demonstration data. These results significantly surpass not only the prior best agents, including AppAgent with GPT-4V (8.3% success rate) and the 17B CogAgent trained with AitW data (38.5%), but also the prior best autonomous RL approach based on filtered behavior cloning (57.8%), thereby establishing a new state-of-the-art for digital agents for in-the-wild device control.

## 1 Introduction

Advances in vision-language models (VLMs), especially in regards to their remarkable common-sense, reasoning, and generalization abilities imply that realizing a fully autonomous digital AI assistant, that can simplify human life by automating day-to-day activities on computer devices via natural language interfaces, is no longer a distant aspiration [16, 45, 56]. An effective device-control AI assistant should be able to complete tasks in-the-wild through Graphical User Interfaces (GUIs) on digital devices: make travel plans; experiment with presentation designs; and operate a mobile device autonomously, all while running amidst stochasticity and distractors on the device, the Internet, and the tools it interacts with. However, enhanced reasoning or common-sense abilities do not directly transfer to intelligent assistant behavior: ultimately we want AI assistants to accomplish

---

*Equal contribution, listed in alphabetical order; work done at UC Berkeley. E-mails: haob2@illinois.edu, yifei_zhou@berkeley.edu, aviralku@andrew.cmu.edu. Project page: https://digirl-agent.github.io/. Code available at https://github.com/DigiRL-agent/digirl.

tasks, exhibit rational behavior, and recover from their mistakes as opposed to simply producing a plausible completion to a given observation based on the data seen during pre-training. This implies that a mechanism to channel abilities from pre-training into a deployable AI "agent" is lacking.

Even the strongest proprietary VLMs, such as GPT-4V [24] and Gemini 1.5 Pro [7] [2], still struggle to produce the right actions when completing tasks on devices. While general-purpose vision-language abilities help these models still make meaningful abstract deductions about novel scenes when deployed, these deductions do not transfer to accurate reasoning for control [47, 45, 55, 44]. As a result, most prior work for building device agents construct complex wrappers around proprietary VLMs by combining them with prompting, search, or tool use [47, 44, 52, 51, 45]. While building prompting or retrieval wrappers to improve decision-making performance of existing VLMs enhances their performance in the short run, without updating the weights, the effectiveness of the resulting agent is inherently limited by the capabilities of the base model [49, 3]. For example, we found that off-the-shelf VLMs make reasoning failures that derail the agent (e.g., Figure 2 and Figure 17), as direct consequences of inability of the base model to reason with low-level device-control actions. A different solution is to fine-tune the model on demonstrations via imitation learning. However, the dynamic nature of the web and device means that models trained to mimic actions in stale data can result in sub-optimalilty as the eco-system changes [26]. Agents trained in this way struggle to recover from the agents' own mistakes [8, 12].

If we can instead build an interactive approach to *train* a VLM to directly adapt and learn *from its own experience* on the device and the Internet, that can be used to build a robust and reliable device-control agent, without needing wrappers on top of proprietary models. However, this learning-based approach must satisfy some desiderata. First, it must make use of online interaction data since static demonstration data would not be representative of the task when the model is deployed: for instance, even in the setting of web navigation alone, dynamic nature of in-the-wild websites means that the agent will frequently encounter website versions that differ significantly from the sce-

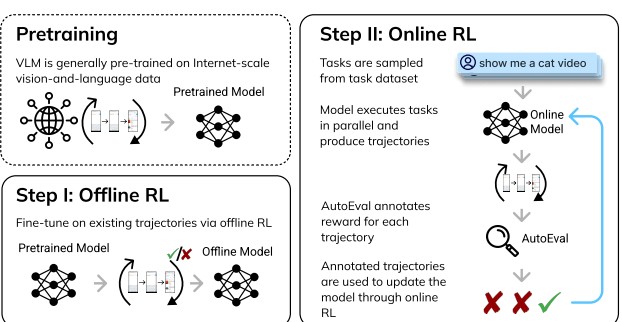

Figure 1: **DigiRL overview.** DigiRL is built upon a VLM that has been pre-trained on extensive web data to develop fundamental skills such as common knowledge, reasoning, and visual grounding. Initially, we employ offline RL to fine-tune the VLM using stale task-specific data, which helps in eliciting goal-oriented behaviors. Subsequently, our agent engages with real-world graphical user interfaces, continuously enhancing its performance through online RL and autonomous performance evaluations.

narios seen during training and will need to behave reliably despite changes in visual appearance and distractions. Second, learning on-the-fly means the approach must learn from multi-turn interaction data from the model itself, a large of chunk of which would consist of failures. Proper mechanisms must be designed to automatically pick out the correct actions while filtering the wrong ones.

To this end, **our main contribution** is a novel autonomous RL approach, DigiRL (i.e., RL for Digital Agents), for training device control agents, as shown in Figure 1. The resulting agent attains state-of-the-art performance on a number of Android device-control tasks. To train this agent, our approach operates in two phases: an initial offline RL phase to initialize the agent using existing data, followed by an offline-to-online RL phase, that further fine-tunes the model obtained from offline RL on online rollout data. Online RL training requires access to an environment that the agent can interact with and obtain reliable reward signals, all in a reasonable amount of wall-clock time. To do so, we build a scalable and parallelizable Android learning environment equipped with a robust VLM-based general-purpose evaluator [26] (average error rate 2.8% against human judgement) that supports running up to 64 real Android emulators at the same time to make online RL real-time. Then, to effectively learn autonomously, we develop an online RL approach that retains the simplicity of supervised learning, but incorporates several key deep RL insights to enable fast fine-tuning. Concretely, our approach is a variant of advantage-weighted regression (AWR) [28], equipped with: **(i)** an automatic curriculum that uses an instruction-level value function to order tasks so as to extract

---

[2]We use external versions of these models as of June 11, 2024. Experiments with GPT and Gemini models were performed entirely by Hao Bai, Yifei Zhou, Mert Cemri, and Jiayi Pan.

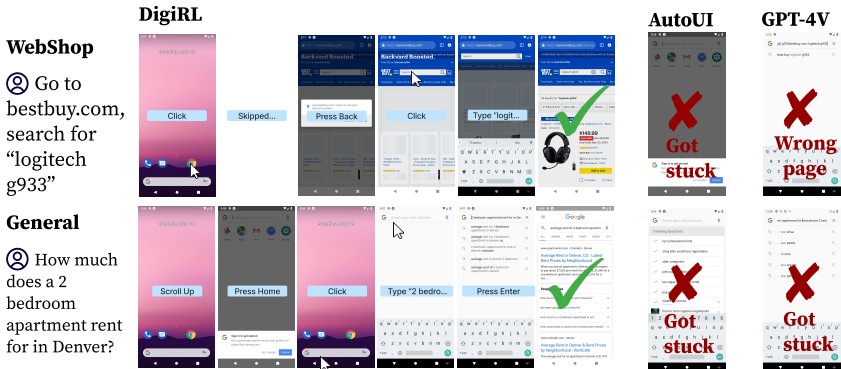

Figure 2: **Qualitative comparison between DigiRL and other approaches.** AutoUI trained from static human demonstrations can easily get stuck in out-of-distribution states while GPT-4V often get on a wrong goal (searched "logitech g933bestbuy.com logitech g933" in Google instead of bestbuy.com). In contrast, DigiRL can recover from such states and complete complex instruction as requested.

maximal learning signal, which is inspired by prioritized replay methods [11, 32, 23], and **(ii)** another step-level value function trained via effective cross-entropy loss [17, 5] to extract low-variance and less-biased learning signal amidst stochasticity and diverse tasks. This RL approach allows us to fine-tune VLMs on their own experience.

We evaluate our agent trained with DigiRL in carrying out diverse instructions from **Android in the Wild dataset** [31] on real Android device emulators and find that our agent can achieve a **28.7% improvement** over the existing state-of-the-art agents (from 38.5% to 67.2% success rate) 18B CogAgent [9], and over 9% improvement over the prior best autonomous learning approach based on Filtered Behavior Cloning [18, 26]. The performance of our agent also significantly surpasses wrappers on top of state-of-the-art proprietary VLMs such as GPT-4V [24] and Gemini 1.5 Pro [7] (17.7% success rate), despite using a significantly smaller model (with 1.3B parameters). To our knowledge, *this is the **first** work to successfully build an autonomous offline-to-online RL approach to enable state-of-the-art performance on device-control problems.*

## 2 Related Work

**Multi-modal digital agents.** In contrast to language-only agents that largely interact with both text or code inputs and outputs [33, 49, 3, 30, 46, 20, 13], training multi-modal agents capable of controlling devices presents different challenges: first, device control is done directly at the pixel-level and in a coordinate-based action space, instead of natural language [31, 44] that LLM is most familiar with, and second, the ecosystem of a device and the Internet tends to be quite stochastic and unpredictable, which is absent with high-level planning in language only. To handle these challenges, prior work largely builds on strong proprietary VLMs [24, 7], and designs complex rule-based wrappers [47, 51, 45, 52] to enhance the visual grounding capabilities of VLMs in GUI interfaces and convert text output into pixel interactions. However, without any form of fine-tuning, this limits the room for possible performance improvement [44, 47, 49, 3, 50], especially when pre-training corpora only present limited action-labeled data. A separate line of work fine-tunes VLMs with demonstration data [19, 15, 9, 53] via imitation learning, but maximizing single-step accuracy from stale demonstrations without accounting for consequences of these actions in subsequent steps may lead to poor solutions amidst stochasticity [26], as agents trained in such ways will struggle to recover from out-of-distribution states not included in the demonstration data [8, 12]. The third category, and perhaps the closest to us, are works that run filtered imitation learning on autonomously-collected data to directly maximize the episode success rate [26, 18]. In contrast, *ours is the first work to scale autonomous, offline-to-online RL* for device control, producing an agent that outperforms prior agents built via imitation. Even when compared to prior work running on-policy RL in simplified web navigation settings (MiniWob++ [37, 10]), our approach is 1000x more sample efficient (around 1e3 trajectories compared to around 1e6 trajectories), and operates in real-world GUI navigation tasks.

**Environments for device control agents.** Recent works have introduced simulated environments for building device control agents [48, 56, 16, 54, 4, 44]. However, these environments are primarily designed for evaluation, and present only a limited range of tasks within fully deterministic and

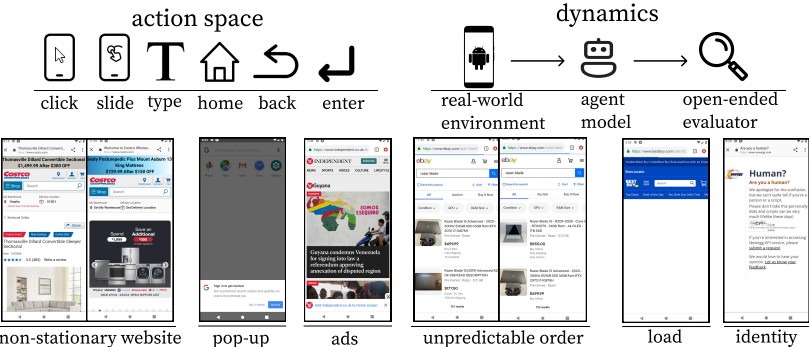

Figure 3: **Environment details.** *Top:* actions space and dynamics of the environment. *Bottom:* examples of the read-world non-stationarity and dynamism of the environment.

stationary settings, infeasible for acquiring a diverse repertoire of skills needed for device control. Alternatively, other works use environments with a greater diversity of tasks [48, 37], but these environments often oversimplify the task complexity, thus failing to transfer to in-the-wild settings. Coversely, our training environment utilizes autonomous evaluation [26] with Gemini 1.5 Pro [7] to support diverse, open-ended tasks on parallel *actual* Android devices, at full scale unlike prior environments. This also contrasts other prior works that use single-threaded Android emulators [26, 39, 19] and thus inefficient for support online RL at scale.

**Reinforcement learning for LLM/VLMs.** The majority of prior research employing RL for foundation models concentrates on tasks that must be solved in a single turn, such as preference optimization [25, 58, 2] or reasoning [27]. However, optimizing for single-turn interaction from expert demonstrations may result in sub-optimal strategies for multi-step problems [57, 38, 42], especially amidst a high degree of stochasticity or non-stationarity. Therefore, we focus on building multi-turn RL algorithms that can learn from sub-optimal, online interaction data in this work. While prior works have developed value-based RL algorithms for LLMs [42, 38, 1, 57, 50], they typically require maintaining multiple models such as Q-networks, value-networks, and policy networks, along with their delayed target counterparts, and can be subjective to slow convergence and sensitivity to choices of hyper-parameters. In contrast, we focus on identifying the key design choices for instantiating a simple yet effective RL algorithm for practitioners to incorporate to substantially improve full-scale Android device control. Our approach can serve as a base model for future research.

# 3 Problem Setup and Preliminaries

**Problem formulation.** We are interested in pixel-based interaction with virtual devices. We scope our study in the control of Android devices: this is already significantly more challenging and more general than previous learning-based environments that focus solely on web navigation [16, 56, 4], where the web browser itself is merely one application within our broader environment, and link-based device controls [47, 51] are inadequate for tasks like games that do not support link inputs.

Each episode begins with the emulator initialized to the home screen. Subsequently, a task is selected from a predefined set of language instructions, some examples of which are shown in Appendix A.1. An agent is then tasked with manipulating the emulator to fulfill this instruction. At each time step, the agent receives a screenshot of the current screen as the observation. Following the action space in prior literature [31], the available actions include tapping and sliding based on normalized $(x, y)$ coordinates (ranging from 0 to 1 relative to the screen dimensions), typing text strings of variable length, and pressing special buttons such as HOME, BACK, and ENTER, as illustrated in Figure 3. Our train and test instructions comes from General and Web Shopping subsets in AitW [31]. These tasks consist of information-gathering tasks like "What's on the menu of In-n-Out?", and shopping tasks on the web like "Go to newegg.com, search for razer kraken, and select the first entry".

**Challenges of stochasticity.** Real-world device contrl presents unique challenges of stochasticity absent in simulated environments [56, 37] such as: **(1)** the non-stationarity of websites and applications, which undergo frequent updates, causing the online observations to be different from stale offline data, **(2)** various unpredictable distractors such as pop-up advertisements, login requests, and the stochastic order of search results. **(3)** technical challenges and glitches such as incomplete webpage loading or temporary access restrictions to certain sites. Examples of scenarios with such stochasticity from

our experiments are shown in Figure 3. We observe that these stochastic elements pose significant challenges for pre-trained VLMs, including even those fine-tuned on device control data. As a concrete example, Figure 4 shows an experiment result that illustrates the necessity of continuously adapting the models to the non-stationarity of websites and applications. After obtaining a good checkpoint using our approach (DigiRL), that we will introduce in the next section, with autonomous data from June.1 to June.3, we compare the performance of a frozen policy and a continuously updating policy using fresh autonomous data from June.7 to June.11. We find that indeed the the performance of the frozen policy gradually degrades over time due to the changes on websites and applications, while continuous online updates plays a key role in preventing this degradation.

**Setup for reliable and scalable online RL.** As autonomous RL interleaves data collection and training, to maximize learning amidst stochasticity, it is crucial to have a real-time data collection pipeline to collect enough experience for gradient updates. While this is not possible in single-thread Android emulator environments [26, 39] due to latency, we parallelize our Android emulator using appropriate error handling as discussed in Appendix A.1. In addition, the environment must provide a reward signal by judging whether the current observation indicates the agent has successfully completed the task. To generalize our *evaluator* to support a wide range of tasks, we extend Pan et al. [26]'s end-to-end autonomous evaluator that does not require accessing the internal states of the emulator or human-written rules for each task. This contrasts previous works that manually write execution functions to verify the functional com-

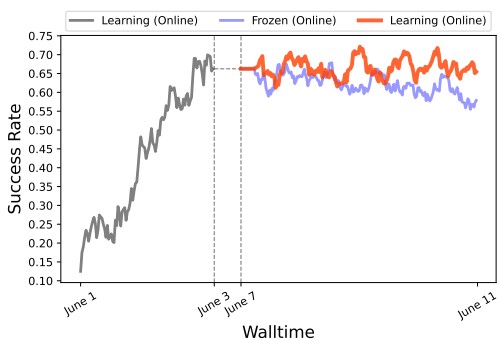

Figure 4: **Performance of our approach (DigiRL) in different training modes** on the Webshop subset. When utilizing a stale checkpoint, i.e., "frozen" (black+blue curve) performance generally begins to degrade as time evolves, whereas autonomous online training (black+red curve) via DigiRL allows us to retain performance despite non-stationarity and stochasticity.

pleteness of each task [16, 48, 37, 44]. We adopt Gemini 1.5 Pro [6, 7] as the backbone of the autonomous evaluator. We seed this model with few-shot rollouts and the associated human-labeled success indicators to guide evaluation of novel queries. This pipeline enables a single evaluator that can evaluate all AiTW tasks. The evaluator is highly aligned with human annotations (average error rate 2.8%), validated in Figure 8.

## 4 DigiRL: Autonomous RL for Building a Strong Device-Control Agent

We now present our autonomous RL framework for training device agents. We pose the device control problem as a Markov decision process (MDP) and develop RL methods for this MDP. The core of our approach is based on a simple and scalable off-policy RL method, advantage-weighted regression (AWR) [29], but we make crucial modifications to handle stochasticity and highly-variable task difficulty, through the use of value functions trained with appropriate losses, and an automatic curriculum, induced by an instruction-level value function to maximize learning.

**Device control and GUI navigation as a MDP.** We conceptualize device control guided by natural language instructions as a finite horizon Markov Decision Process (MDP) represented by $\mathcal{M} = \{\mathcal{S}, \mathcal{A}, \mathcal{T}, \mu_0, \mathcal{R}, H\}$ and run policy gradient to solve this MDP. At the beginning, an initial state $s_0$ and a natural language instruction $c$ are sampled from the initial state distribution $\mu_0$. A reward of 1 is given at the end if the agent successfully fulfills the task per the evaluator, otherwise a reward of 0 is given. The trajectory terminates either when the agent accomplishes the task or when the maximum allowed number of interactions $H$ is exceeded. States are represented using the last two screenshots. To explain our approach in detail, we also include several standard definitions used in reinforcement learning (RL). The Q function for a policy $\pi$ represents the expected long-term return from taking a specific action at the current step and then following policy $\pi$ thereafter: $Q^\pi(s_h, a_h, c) = \mathbb{E}_\pi \left[ \sum_{t=h}^H r(s_t, a_t, c) \right]$. The value function $V^\pi(s_h, c)$ is calculated by averaging the Q-value, $Q^\pi(s_h, a_h, c)$, over actions $a_h$ drawn from the policy $\pi$. The advantage $A^\pi(s_h, a_h, c)$ for a state-action pair is computed by subtracting the state's value under the policy from its Q-value: $A^\pi(s_h, a_h, c) = Q^\pi(s_h, a_h, c) - V^\pi(s_h, c)$.

## 4.1 Backbone of Our Approach: Off-Policy RL via Advantage-Weighted Regression

The starting point we choose to build our approach on is the advantage-weighted regression (AWR) algorithm [29], which says that we can improve the policy reliably by regressing the policy towards exponentiated advantages induced by the reward function, as a proxy for optimizing the policy gradient while staying close to the previous policy [14, 35, 34]:

$$\arg\max_\pi \mathbb{E}_\nu \left[\log \pi(a|s, c) \cdot \exp\left(A(s, a, c)/\beta\right)\right], \qquad (4.1)$$

for some positive parameter $\beta$ and the distribution of past experience $\nu$, and $A(s, a, c)$ denotes the advantage of a state-action pair $(s, a)$ given a context $c$. To avoid tuning the hyperparameter $\beta$, we consider an alternative that does "hard filtering" on the advantages instead of computing $\exp(A)$, similar to prior works [22, 43]. This leads to the following loss function for fine-tuning the model:

$$\mathcal{L}(\pi) = -\mathbb{E}_{\text{filter}(\nu)}[\log \pi(a|s, c)]. \qquad (4.2)$$

Typically, these advantages are computed by running Monte-Carlo (MC) rollouts in the environment to estimate the value of a given state-action pair, and subtracting from it an estimate of the value of the state given by a learned value estimator alone. However, this approach is likely to produce high-variance advantages given the stochasticity of the device eco-system that affects MC rollouts.

## 4.2 Obtaining Reliable Advantage Estimates from Doubly-Robust Estimators

To reliably identify *advantageous* actions given significant environment stochasticity, we construct a per-step advantage estimator, inspired by doubly-robust estimators [40, 36]:

$$A^{\text{step}}(s_h, a_h, c) := \lambda^{H-h} r(s_H, a_H, c) + (1 - \lambda^{H-h} r(s_H, a_H, c))(V^{\text{step}}(s_{h+1}, c) + r(s_h, a_h, c) - V^{\text{step}}(s_h, c)), \qquad (4.3)$$

where $\lambda$ is a weighting hyper-parameter. This construction of the advantage estimator is a simplified version of Generalized Advantage Estimation (GAE) [36] using only the next-step advantage estimator and final-step advantage estimator as there are no intermediate rewards in our problem. This construction balances an advantage estimator with higher variance Monte-Carlo estimates $\lambda^{H-h} r(s_H, a_H, c)$ (due to stochasticity) and an estimator with higher bias $V^{\text{step}}(s_{h+1}, c) + r(s_h, a_h, c) - V^{\text{step}}(s_h, c)$ (due to imperfect fitting of the value function). We observed that combining both high-variance and high-bias estimators gave us a sweet-spot in terms of performance. To implement the step-level hard filtering, we simply threshold this doubly robust estimator as $A^{\text{step}}(s_h, a_h, c) > 1/H$ to decide which actions progress towards the goal.

## 4.3 Automatic Curriculum using an Instruction-Level Value Function

While the AWR update (Equation 4.1) coupled with a robust advantage estimator (Equation 4.3) is likely sufficient on standard RL tasks, we did not find it to be effective enough for device control in preliminary experiments. Often this was the case because the task set presents tasks with highly-variable difficulties that collecting more data on tasks that the agent was already proficient at affected sample efficiency negatively. In contrast, maximal learning signal can be derived by experiencing the most informative tasks for the agent during training. To this end, we design an instruction-level value function $V^{\text{instruct}}(c)$ to evaluate if a given rollout can provide an effective learning signal:

$$A^{\text{instruct}}(s_h, a_h, c) := \sum_{t=h}^{H} r(s_t, a_t, c) - V^{\text{instruct}}(c) = r(s_H, a_H, c) - V^{\text{instruct}}(c), \qquad (4.4)$$

where $\sum_{t=h}^{H} r(s_t, a_t, c)$ is a Monte-Carlo estimator of $Q(s_h, a_h, c)$. The equality holds because the MDP formulation only provides rewards at the end of a rollout. Intuitively, if a rollout attains a high value of $A^{\text{instruct}}(s_h, a_h, c)$, it means the value function $V^{\text{instruct}}$ is small. Therefore, this rollout represents a valuable experience of the agent accomplishing a difficult task, and thus should be prioritized, akin to ideas pertaining to prioritized experience [32] or level replay [11]. When training the actor with a buffer of historical off-policy data, we first perform a filtering step to identify the top-$p$ datapoints with highest $A^{\text{instruct}}(s_h, a_h, c)$. Then, we use it for AWR (Equation 4.1) with the doubly-robust advantage estimator (Equation 4.3).

**Implementation details.** Inspired by the findings in some recent works [5, 17] that modern deep learning architectures like transformers [41] are better trained with cross-entropy losses instead of mean-squared losses, we utilize a cross-entropy objective based on the Monte-Carlo estimate of the trajectory reward for training both of our value functions:

$$\mathcal{L}(V^{\text{traj}}) = -\mathbb{E}_\nu[r(s_H, a_H, c) \log V^{\text{traj}}(c) + (1 - r(s_H, a_H, c)) \log(1 - V^{\text{traj}}(c))], \qquad (4.5)$$

$$\mathcal{L}(V^{\text{step}}) = -\mathbb{E}_\nu[r(s_H, a_H, c) \log V^{\text{step}}(s_h, a_h, c) + (1 - r(s_H, a_H, c)) \log(1 - V^{\text{step}}(s_h, a_h, c))]. \qquad (4.6)$$

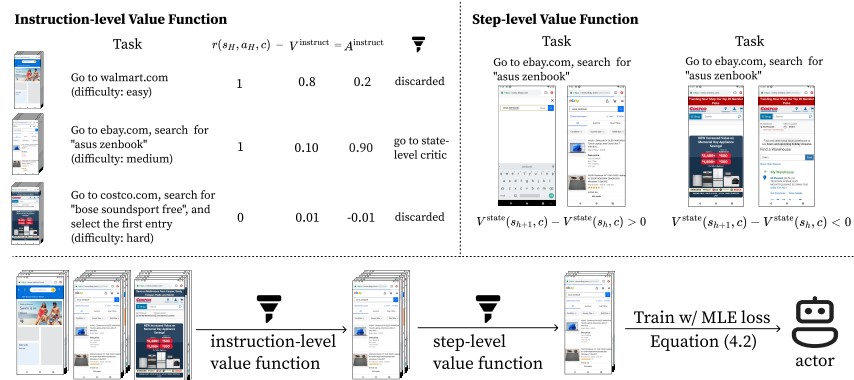

Figure 5: **Algorithm visualization.** The two value function are first trained with original distribution of collected trajectories according to Equation (4.5) and Equation (4.6), then used to filter the trajectories for training the actor. We use the MLE loss (Maximum Likelihood Estimation loss) to train the actor.

**Final algorithm.** The final practical algorithm is shown in Figure 5. The instruction-level value function estimates the values of the trajectories, which is trained with loss shown in Equation (4.5). The step-level value function estimates the values of states, which is trained with loss shown in Equation (4.6). When training the actor, we first filter out trajectories and states using the value functions as shown in Equation (4.4) and Equation (4.3), then train the actor with the MLE loss shown in Equation (4.2) on the filtered data.

## 5 Experimental Evaluation

The goal of our experiments is to evaluate the performance of DigiRL on challenging Android device control problems. Specifically, we are interested in understanding if DigiRL can produce agents that can effectively learn from autonomous interaction, while still being able to utilize offline data for learning. To this end, we perform a comparative analysis of DigiRL against several prior approaches, including state-of-the-art agents in Section 5.1. We also perform several ablation experiments to understand the necessity and sufficiency of various components of our approach in Section 5.2.

**Baselines and comparisons.** We compare DigiRL with: **(a)** state-of-the-art agents built around proprietary VLMs, with the use of several prompting and retrieval-style techniques; **(b)** running imitation learning on static human demonstrations with the same instruction distribution, and **(c)**a filtered BC approach [26]. For proprietary VLMs, we evaluate **GPT-4V** [24] and **Gemini 1.5 Pro** [7] both zero-shot and when augmented with carefully-designed prompts. For the zero-shot setting, we use the prompt from Yang et al. [47] and augment the observation with Set-of-Marks [55]. Set-of-Marks overlays a number for each interactable element over the screenshot, so that a VLM can directly output the number of the element to interact with in plain text instead of attempting to calculate pixel coordinates, which is typically significantly harder. We also compare with AppAgent [47], which first prompts the VLM to explore the environment, and appends the experience collected to the test-time prompt. We also compare with two state-of-the-art fine-tuning methods for Android device control: **AutoUI** (specifically AutoUI-Base [53]) and **CogAgent** [9]. AutoUI-Base uses an LM with 200M parameters, and a a vision encoder with 1.1B parameters. CogAgent has 11B parameters for its vision encoder and 7B for its LM. The supervised training corpus for both AutoUI-Base and CogAgent contains AitW, including the instruction set and the emulator configuration we use.

**Base VLM and offline dataset.** Both **Filtered BC** and **DigiRL** use trained AutoUI-Base checkpoints with the image encoder frozen. The instruction and step-level value functions for DigiRL employ this same frozen image encoder. The visual features output from the encoder are concatenated with instruction features derived from RoBERTa [21]. A two-layer MLP is then used to predict the value function. In the offline phase, the offline dataset is collected by rolling out the initial AutoUI-Base supervised trained checkpoint as policy. For fair comparisons, we keep the number of offline data collected in the pure offline training roughly the same as the total number of data collected in the offline-to-online training. Due to the dynamic nature of the Internet-device eco-system, our offline data was stale by the time we were able to run our offline-to-online experiments, and this presented additional challenge in offline-to-online learning. In both General and Web Shopping subsets, offline experiments make use of around 1500 trajectories while offline-to-online experiments start with

|  |  |  | AitW General | | AitW Web Shopping | |
|---|---|---|---|---|---|---|
|  |  |  | Train | Test | Train | Test |
| **Prompting** | SET-OF-MARKS | GPT-4V | 5.2 | 13.5 | 3.1 | 8.3 |
|  |  | Gemini 1.5 Pro | 32.3 | 16.7 | 6.3 | 11.5 |
|  | APPAGENT | GPT-4V | 13.5 | 17.7 | 12.5 | 8.3 |
|  |  | Gemini 1.5 Pro | 14.6 | 16.7 | 5.2 | 8.3 |
| **Learning** | SUPERVISED TRAINING | CogAgent | 25.0 | 25.0 | 31.3 | 38.5 |
|  |  | AutoUI | 12.5 | 14.6 | 14.6 | 17.7 |
|  | OFFLINE | Filtered BC | $51.7 \pm 5.4$ | $50.7 \pm 1.8$ | $44.7 \pm 1.6$ | $45.8 \pm 0.9$ |
|  |  | **Ours** | $46.9 \pm 5.6$ | $62.8 \pm 1.0$ | $39.3 \pm 6.0$ | $45.8 \pm 6.6$ |
|  | OFF-TO-ON | Filtered BC | $53.5 \pm 0.8$ | $61.5 \pm 1.1$ | $53.6 \pm 4.7$ | $57.8 \pm 2.6$ |
|  |  | **Ours** | $\mathbf{63.5} \pm \mathbf{0.0}$ | $\mathbf{71.9} \pm \mathbf{1.1}$ | $\mathbf{68.2} \pm \mathbf{6.8}$ | $\mathbf{67.2} \pm \mathbf{1.5}$ |

Table 1: **Main comparisons of different agents across various settings.** Each offline experiment is repeated three times and the mean and standard deviation are reported. Each online experiment is repeated two times. Results are evaluated with our autonomous evaluator with the first 96 instructions in the train and test set. Correlation of our correlation and human judgements can be found in Figure 8.

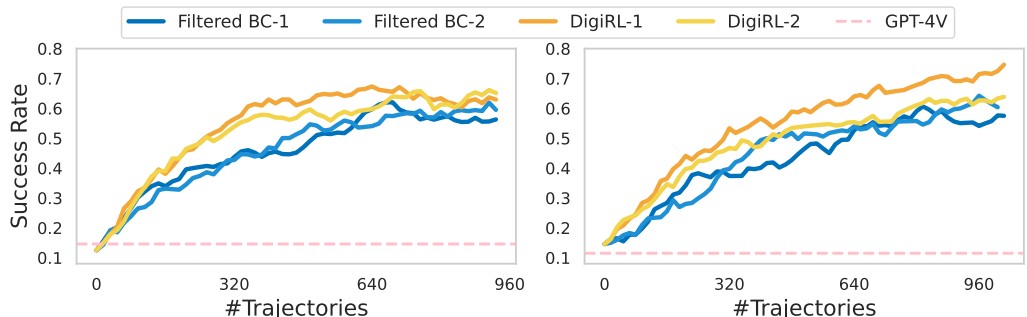

Figure 6: **Offline-to-online training curves for Filtered BC and DigiRL.** Curves are smoothed with exponential weighting over the x-axis. *Left:* AitW General. *Right:* AitW Web Shopping. Two runs for each model are started on two different dates with at least two days apart. Observe that DigiRL is able to improve faster with a fewer number of samples. Since the data collection frequency is the bottleneck, these performance trends directly reflect performance trends against wall-clock time as well.

around 500 offline trajectories and update with another 1000 online trajectories. In the offline phase, DigiRL skips instruction-level filtering and instead trains the actor with all successful trajectories to make full use of the offline data. See a detailed breakdown of our dataset in Appendix A.1.

## 5.1 Main Results

Our main results are summarized in Table 1 and Figure 6. We find that on both AitW General and AitW Web Shopping subsets, the agent trained via DigiRL significantly outperforms prior state-of-the-art methods based on prompting and retrieval (AppAgent + GPT-4V/Gemini 1.5 Pro) or training on static demonstrations (CogAgent and AutoUI), a large margin with more than **49.5% absolute improvement** (from 17.7% to 71.9% on the General subset and from 17.7% to 67.2% on the Web Shopping subset). Notably, this improvement from DigiRL is realized *fully autonomously without making use of human supervision* (e.g. manually labeled rollouts or hand-written verifiers).

**Are inference-time prompting and retrieval techniques or supervised training enough for device control?** Delving into Table 1, we observe that off-the-shelf proprietary VLMs, even when supplemented with the set-of-marks mechanism, do not attain satisfactory performance: both GPT-4V and Gemini 1.5 Pro achieve success rates under 20%. One possible cause could be the under-representation of Android device data in the pre-training data. Moreover, inference-time adaptation strategies such as AppAgent [47] show minimal improvement, with gains not exceeding 5% for either model. All this evidence suggests a limited scope for improvement without fine-tuning of some sort. As illustrated in Figure 7, the primary failures of these VLMs stem from hallucinatory reasoning that lead the VLMs to land on a relevant but wrong page. This suggests that while state-of-the-art VLMs excel at reasoning problems in code and math, their reliability in less-familiar domains, such

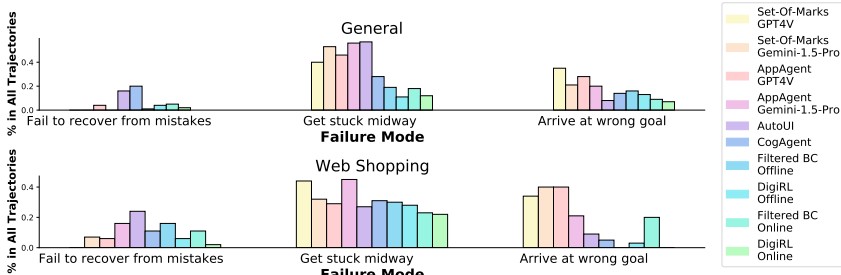

Figure 7: **Failure modes for each approach** on both the AiTW General and Web Shopping subsets. We found that the failure mode RL training is most effective at reducing compared to model supervised trained on human data is "Fail to recover from mistakes". A more fine-grained decomposition can be found in Appendix D.

as device control, remains inadequate. For example, for the instruction "Go to newegg.com, search for alienware area 51, and select the first entry", a GPT-4V based agent erroneously searched "alien area 51 ebay" in Google.com and decided that it had made progress towards the task (Figure 17).

Training on domain-specific human demonstrations, however, does boost performance, allowing the smaller, specialized VLM, AutoUI with 1.5 billion parameters, to match or surpass the larger, generalist VLMs like GPT-4V and Gemini 1.5 Pro. Nonetheless, this supervised imitation learning approach still fall short, with success rates on both subsets remaining below 20%. This shortcoming is not fundamentally addressed via enhancements in model scale or architecture, as evidenced by CogAgent [9], with 18 billion parameters still achieving performances below 40% success rate. As depicted in Figure 7, a predominant failure mode for these agents is an inability to rectify their own errors. An example trajectory that we observed is that for the instruction "what's on the menu of In-n-Out", the agent accidentally activated the voice input button, and failed to quit that page until the step limit. In contrast, DigiRL is able to recover from the errors more efficiently( Appendix C.2).

**Comparison of different RL approaches.** In Table 1 and Figure 6, we present a comparative analysis of various autonomous approaches. Notably, both offline and offline-to-online configurations demonstrate that our RL approach, when augmented with a continuous stream of autonomous interaction data and reward feedback, substantially improves performance. This improvement is evident from an increase in the success rate from under 20% to over 40%, as the agent learns to adapt to stochastic and non-stationary device interfaces. Moreover, although the total sample sizes for offline and offline-to-online settings are equivalent, the top-performing offline-to-online algorithm markedly surpasses its offline counterpart (75% versus 62.8% on the General subset). This highlights the efficacy of autonomous environment interaction, and establishes the efficacy of DigiRL in learning from such uncurated, sub-optimal data. Lastly, DigiRL consistently outperforms the state-of-the-art alternative, Filtered BC, across both the General and Web Shopping subsets, improving from 61.5% to 71.9% and 57.8% to 61.4%, respectively, highlighting DigiRL's performance and efficiency.

### 5.2 Analysis and Ablations

**Failure modes analysis.** We conduct an additional user study to annotate the failure modes for each agent as shown in Figure 7, and a more fine-grained breakdown can be found in Appendix D. At a high level, we classify the major failure modes of all agents into the following three categories: **(1)** *Failure to recover from mistakes* refers to the scenario where the agent made a mistake that led it to states from which it failed to quickly recover and resume the task, such as a wrong search page. **(2)** *Getting stuck midway* refers to the failure mode where the agent gets distracted on the right track to completing the instruction and as a result fails to accomplish the task. For example, failing to click on the right link or failing to search after typing the key words. **(3)** *Arriving at wrong goal* refers to the failure mode where the agent arrives at a wrong page and mistakenly thinks that it had completed the task. For e.g, the agent finds a macbook on costco.com instead of finding a macbook on ebay.com.

While all the types of failure modes benefit from offline and offline-to-online RL training as shown in Figure 7, the most consistent and significant reduction is probably for the failure mode of failing to recover from mistakes. This is because while pre-trained models, generating plausible future tokens, can get distracted by the dynamic nature of the environment and, as a result, encounter at never-before-seen states. With no clue of how to escape such states, these methods are unable to recover and fail to solve the task. In contrast, by training on autonomously-collected rollouts, our agent DigiRL is able to learn from its own mistakes and reduces failures to recover over training.

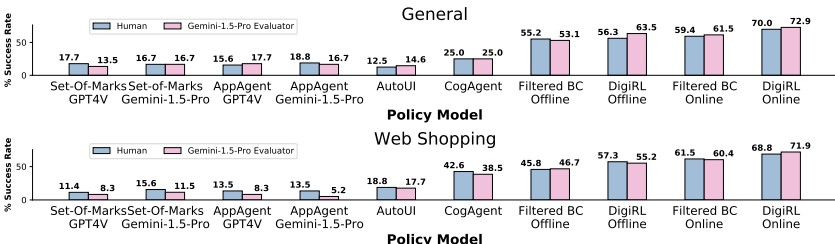

Figure 8: **Correlation between our autonomous evaluator and human judgements for all policy models** on General and Web Shopping subsets. For repeated offline and online runs, we report the correlation results for the run with the highest autonomous evaluation success rate.

**Ablation study of each component in DigiRL.** We conduct an ablation study on different components of DigiRL in Figure 9. We find that all the components used by our approach are necessary: **(1)** using cross-entropy for training the value functions boosts performance by around 12% (compare Ours and Ours w/ Regression); **(2)** using step-level advantages improves efficiency by 12% (comparing Ours and Ours w/o step-level advantage); **(3)** the use of automatic curriculum improves the speed of learning by around 25% (comparing Ours w/o step-level advantage and Filtered BC); **(4)** Ours outperforms vanilla AWR that does not employ a doubly-robust advantage estimator or curriculum.

Additionally, we also observe no degradation in performance as a result of "hard-filtering", as show by nearly comparable performance of our approach and the best run of exponential filtering obtained via an extensive tuning of the temperature hyperparameter $\tau$ in naïve AWR (comparing Ours and Ours w/ vanilla AWR reweighting), despite simplicity of implementation in the hard filtering approach. Putting together, these choices result in a new state-of-the-art RL approach for device control.

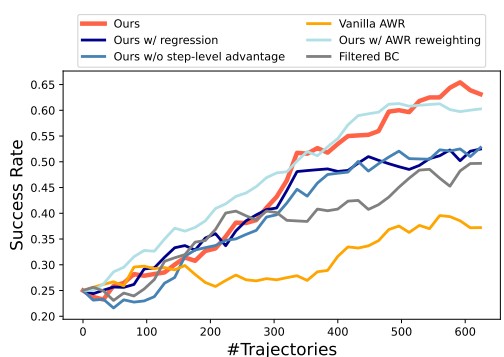

Figure 9: **Ablation study results on the AitW Web Shopping subset.**

**Evaluation of our autonomous evaluator.** In Figure 8, we present the findings from a user study aimed at assessing the accuracy of our autonomous evaluator. Our results indicate that the success rates reported by our automatic evaluator are remarkably consistent with those assessed by human evaluators across almost all models, with differences less than 3%. Furthermore, we observed that evaluations on the Web Shopping subset are more precise compared to those on the General subset. This increased accuracy likely stems from the fact that tasks in the General subset are formulated in free-form language, which can introduce ambiguity, whereas the Web Shopping subset features a narrower range of language expressions, reducing potential variability.

## 6  Discussion and Limitations

In this paper, we propose a novel autonomous RL approach, DigiRL, for training in-the-wild, multi-modal, device-control agents that establish a new state-of-the-art performance on a number of Android control tasks from Android-in-the-Wild dataset [31]. To achieve this, we first build a scalable and parallelizable Android environment with a robust VLM-based general-purpose evaluator that supports fast online data collection. We then develop a system for offline RL pre-training, followed by autonomous RL fine-tuning to learn via interaction, admist the stochasticity of the real-world Internet and device eco-system. Our agent achieves a 280% improvement over the previous state-of-the-art agents (from 17.7% to 68.2% in terms of task success rate), including AppAgent based on GPT-4V and Gemini 1.5 Pro, and supervised trained models such as AutoUI and CogAgent.

Due to computational limitations, and despite the fact that the parallel emulator and autonomous evaluator can be easily extended to complicated tasks, our agent is trained only with tasks from AitW instead of a all possible tasks on the device. Our design of the DigiRL algorithm aims for maximal implementation simplicity, so we hope that our approach to serve as a base algorithm for future research to build on, including algorithmic research as well as expanding the space of tasks.

## Acknowledgements

We thank Yi Su, Izzedin Gur, Xinyang Geng, and Sandra Faust for feedback on an earlier version of this paper and for informative discussions. This work is supported by NSF IIS-2246811 and ONR N00014-21-1-2838, and Gemini 1.5 Pro credit donations for academic use and cloud resources from Google Cloud.

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

# Appendices

## A    Environment details

### A.1    Post-processing of AitW

The Android in the Wild (AiTW) task set is a large-scale dataset for android device control, containing five subsets: GoogleApps, Install, Web Shopping, General, and Single, where we select the General and Web Shopping subsets. Single subset is not considered here because all tasks in Single can be completed within one step and thus this subset fails to examine the multi-step challenges that we are interested in this paper. Install and GoogleApps are not considered due to security reasons as those tasks require an active Google account and parallel emulations can flag security concerns.

**General.** The General set focuses on searching for information and basic application usage. For example, it contains searching for latest news in Chile, search for flights from NYC to Sydney, opening Gmail, etc. We use all 545 tasks in the training set for training and the first 96 tasks in the test set for testing due to computational and budget constraints. The maximum allowed number of steps for this subset is 10. Offline data is collected by rolling our the initial AutoUI policy with tasks from the training set. The offline data used for the offline-to-online setting contains 608 trajectories while the offline data used for the offline setting contains 1552 trajectories. Some task examples are shown in Table 3.

| Task Example |
| --- |
| How do I get to the nearest Verizon Store? |
| How much does a 2 bedroom apartment rent for in Denver? |
| Search for flights from Barcelona to Boston |
| What's a good restaurant in New York? |
| What's on the menu at Burger King? |

Table 2: Examples of task descriptions in the AiTW General task set.

**Web Shopping.** The Web Shopping subset comprises search instructions on various shopping websites, like searching for razer blader on ebay. As some websites (e.g. Amazon) and operations (e.g. adding items to cart) frequently require captcha verifications, we post-process the Web Shopping subset to exclude such operations and websites and also make the task easy to evaluate for our autonomous evaluator. The resulting task set involves navigating through five websites (costco.com, bestbuy.com, target.com, walmart.com, newegg.com) and three basic operations (go to website, search in the website, and select items from the searched results). Our post-processed training set contains 438 tasks and our testing set contains 96 tasks. Example tasks after post-processing can be found in Table 3. The maximum allowed number of steps for this subset is 20. Offline data is collected by rolling our the initial AutoUI policy with tasks from the training set. The offline data used for the offline-to-online setting contains 528 trajectories while the offline data used for the offline setting contains 1296 trajectories.

| Difficulty | Task Example |
| --- | --- |
| 1 | Go to costco.com |
|  | Go to walmart.com |
| 2 | Go to costco.com, search for "bose soundsport free" |
|  | Go to walmart.com, search for "logitech g910" |
| 3 | Go to costco.com, search for "bose soundsport free" and select the first entry |
|  | Go to walmart.com, search for "logitech g910" and select the first entry |

Table 3: Examples of task descriptions in the AiTW Webshopping task set.

|  | AitW General | | AitW Web Shopping | |
|---|---|---|---|---|
|  | All Trajectories | Successful Trajectories | All Trajectories | Successful Trajectories |
| DigiRL Run1 | 6.31 | 4.40 | 11.35 | 7.23 |
| DigiRL Run2 | 6.64 | 5.04 | 10.86 | 6.55 |
| Filtered BC Run1 | 8.08 | 6.56 | 12.05 | 6.88 |
| Filtered BC Run2 | 7.36 | 6.13 | 14.72 | 9.62 |

Table 4: **Average rollout length of the DigiRL agent compared to filtered BC.** Darker green means shorter rollout length. On both AitW General and AitW Web Shopping test subsets, we find that DigiRL consistently produces shorter length rollouts than filtered BC.

# B  Other Quantitative Experiments

## B.1  Curriculum Learning

When running experiments on the AitW Web Shopping subset, we find solving easier tasks helps improve solving harder tasks, where the difficulty is identified in Table 3. By specifying the difficulty DigiRL-Run1 in Figure 6, we empirically show the success rates of each difficulty across the online learning process in Figure 12, we observe that a significant increase of success rate of tasks of difficulty 1 leads to increasing success rate of difficulty 2, and the same pattern for difficulty 2 and 3, demonstrating effective curriculum learning.

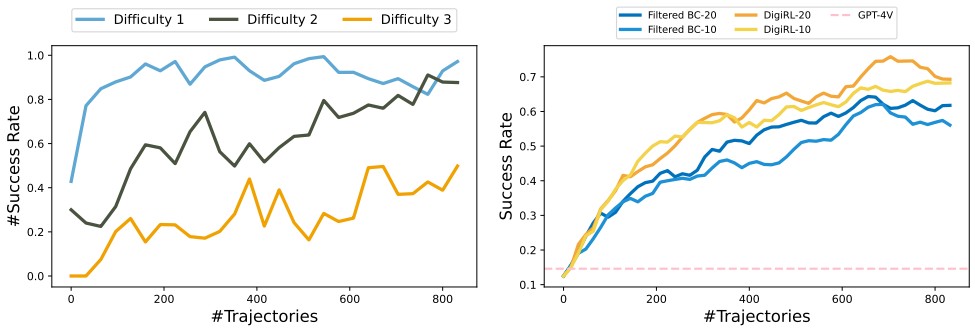

Figure 10: *Left:* Success rate under different difficulties for the AiTW Webshopping task set. *Right:* Success rate under different methods with different horizon length ($H \in \{10, 20\}$) on the AiTW Google Search task set.

## B.2  Learning Method

We ablate on the learning method, i.e. online learning or offline-to-online learning. We find that offline-to-online learning converges faster than online learning, and is not necessarily worse than online learning in terms of final performance, as shown in Figure 11.

## B.3  Horizon Limit

We investigate the horizon limit of filtered BC and DigiRL on the AitW General subset. As most tasks can be effectively solved within 10 steps, we specify two horizon limits: a sufficient horizon $H = 10$, and a redundant horizon $H = 20$. Results in Figure 12 show that a redundant horizon introduces significantly faster learning speed for both filtered BC and DigiRL, presumbaly because longer horizon means more opportunity to try in a single trajectory. In both horizon settings, we observe the DigiRL offers a significant speedup of around 100 trajectories over Filtered BC.

## B.4  Trajectory Length

We investigate the rollout length of DigiRL compared to filtered BC. Results in Table 4 demonstrate that DigiRL consistently achieves shorter average rollout lengths compared to filtered BC across both

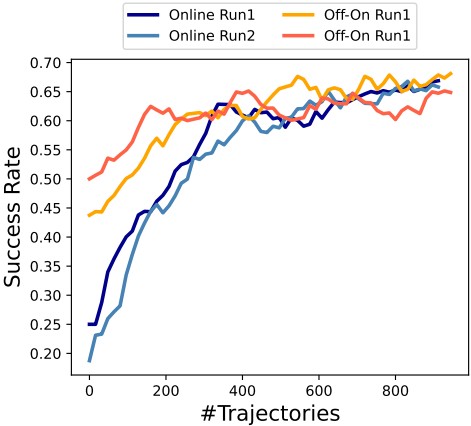

Figure 11: **Success rate with pure online learning or offline-to-online learning** w.r.t. the number of online trajectories trained on the AitW General dataset. The starting points of curves in this figure look different from the main results figure because the starting points of the main results figure is smoothed at the average performance of the offline trajectories collected for the offline-to-online learning.

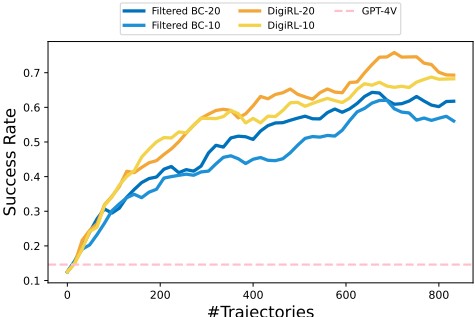

Figure 12: **Success rate with different horizon length** ($H \in \{10, 20\}$)under different methods on the AiTW Google Search task set.

subsets. This observation holds true whether considering all rollouts for computing this correlation or only investigating this correlation on rollouts that eventually succeed. This indicates the capability of DigiRL to solve tasks in a more efficient and directed manner. Qualitative examples can be found in Figure 16.

# C    Qualitative Examples

## C.1    Random sample of trajectories for different agents

In Figures 13 and 14, we provide trajectories of DigiRL, AutoUI, and GPT-4V randomly sampled from our test set to offer a qualitative understanding of the agents' performance. As shown in these examples, DigiRLcan efficiently carry out in-the-wild device control tasks and less likely to get stuck or get to a wrong page compared to AutoUI and GPT-4V.

## C.2    Error Recovery

We observe that DigiRL is able to recover from its own mistakes. As shown in Figure 15, we find that DigiRL explores ways to get back to the original screen in order to perform a search. As a comparison, AutoUI fails to reset to the original screen and gets stuck at the diverged screen. Under the hood, we find DigiRL trying to maximize the state value, which usually induces it to reset to the original screen (that has a large value to success).

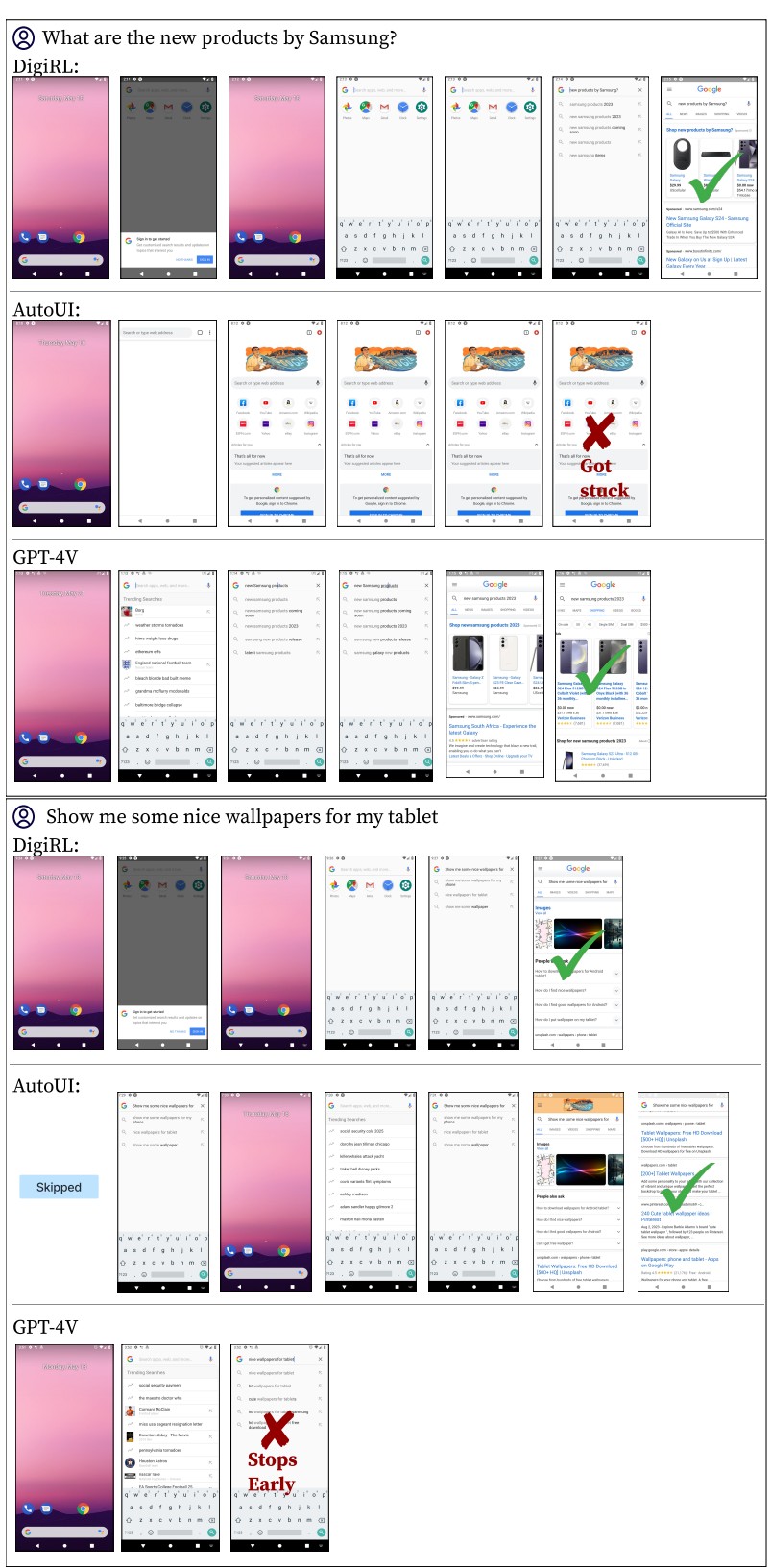

Figure 13: Agents' trajectory on two randomly sampled tasks on the General split of AitW.

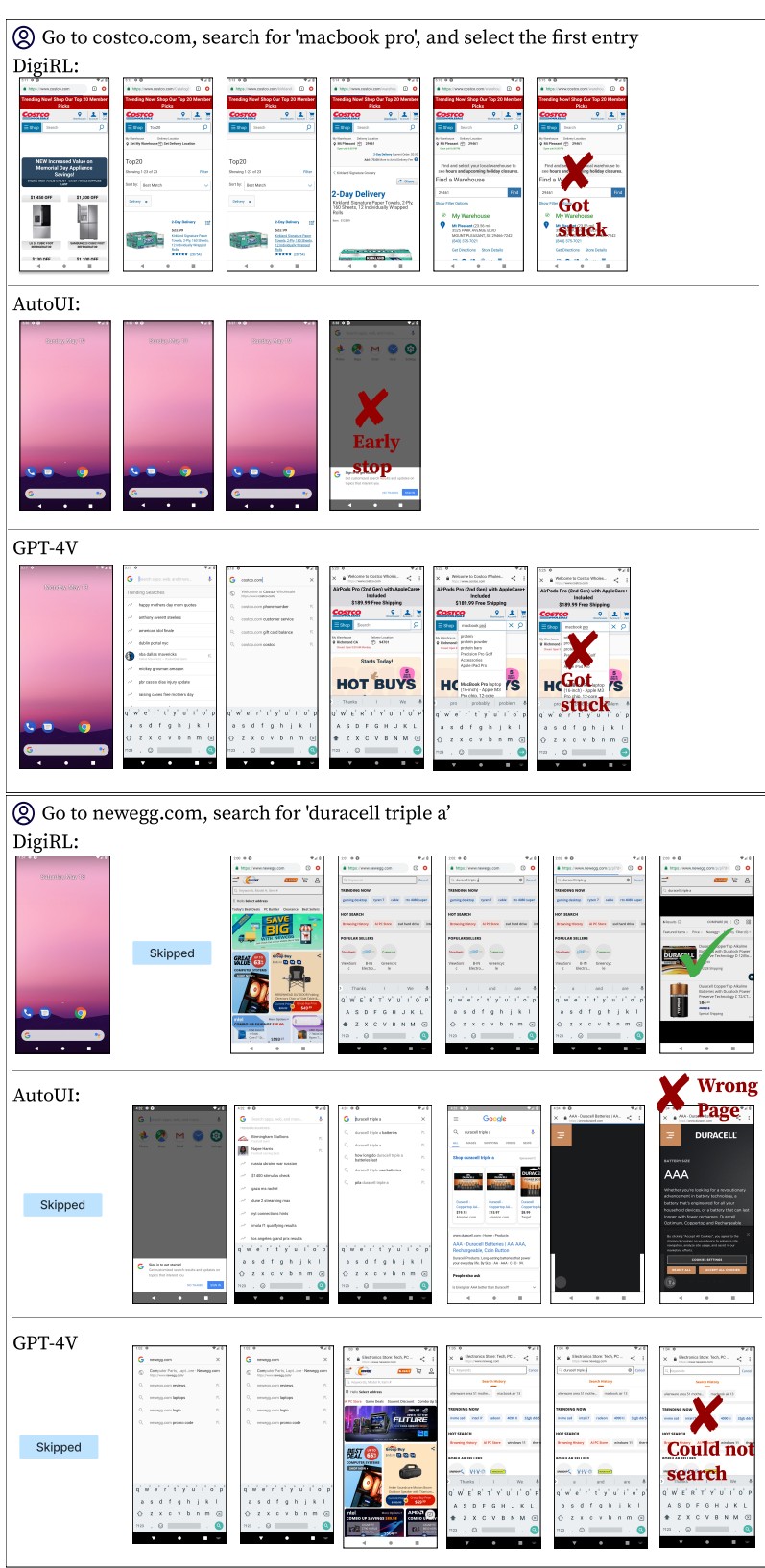

Figure 14: Agents' trajectory on two randomly sampled tasks on the WebShop split of AitW.

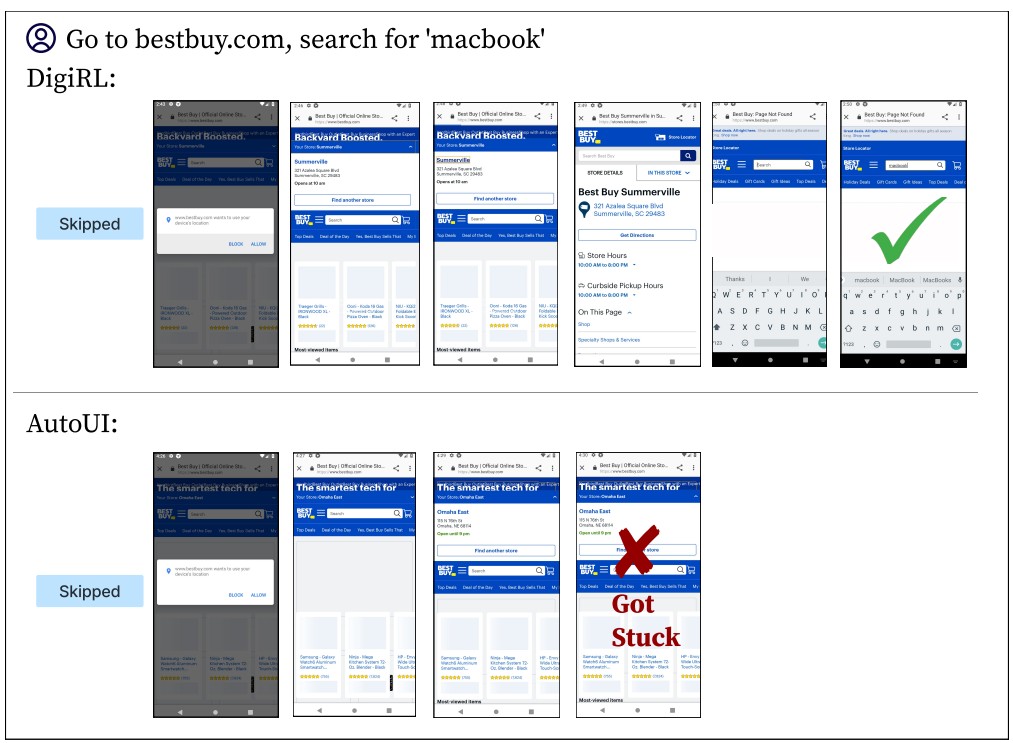

Figure 15: Error recovery cases. In `bestbuy.com`, we systematically find DigiRL able to recover from its own mistakes, while AutoUI fails to do so.

### C.3  Trajectory Length

Qualitative example on the number of steps in trajectories of DigiRL and filtered BC are shown in Figure 16. We find consistent cases where DigiRL has shorter trajectory length than filtere BC.

### C.4  Reasoning failure of GPT-4V

The performance of GPT-4V failed on AiTW tasks predominantly due to not being able to carry out control actions as it plans on a high level, and then not being able to recover from these mistakes. Moreover, one of the main reasons why it is not able to recover from a mistake is that it might hallucinate and make itself believe that it is a wrong app or website. Indeed, GPT-4V constructs a plan of further actions when provided a task from either Web Shopping or General dataset of AiTW. Then, when it makes a misclick and fails to successfully proceed in an intermediate step, it might think that it actually solved that intermediate step and is in the correct app or website to execute further actions, causing the overall trajectory to fail. An example of this is provided in Figure 17. Here, we ask the model to search for an item in a webshopping website, in particular in "newegg.com". However, the model fails to proceed to that website due to not being able to precisely locating the search button. Then, instead of trying to go to that website again, the model thinks it is already in that webshopping website, and mistakes the search bar of Google with the search bar of "newegg.com". Hence, the rest of the trajectory also fails. Another slightly different phenomenon is illustrated in Figure 18. Here, the model is able to proceed to the correct website and search for an item, but this time it fails to tap on the search button on the website and clicks to an advertisement instead. Consequently, the model fools itself to think it successfully searched the item, and scrolls the page hoping to find that item, but it cannot do so because in reality it views the results of the advertisement. The primary reason of these failures is the challenge of grounding the control actions in GUI interfaces to realize the intermediary goals laid out by GPT-4V model's thoughts. As an example, we provide an illustration of trying to set up an alarm task in Figure 19. Here, in the last frame, it fails to execute the precise movements in the necessary amount of rounds to correctly set up the alarm to the desired time, and in the last frame we see that the action taken does not align with the thought process of the model.

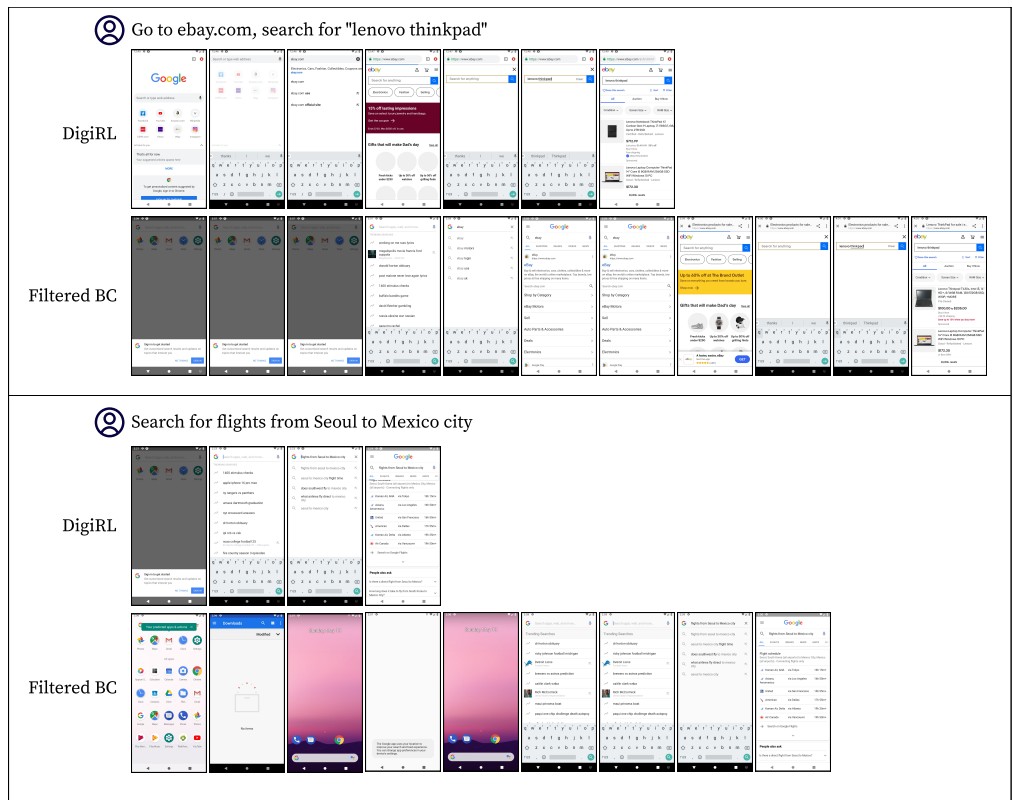

Figure 16: Examples where DigiRL has shorter trajectory length than online filtered BC.

## D   Fine-grained failure modes

In Figure 20, we present a more fine-grained breakdown for all six failure modes provided in the user study. Those failure modes include:

- *Failure to recover from mistakes* refers to the scenario where the agent made a mistake that led it to states from which it failed to quickly recover and resume the task, such as a wrong google search page.

- *Failure to click on the right link or failure to click* refers to the failure mode where the agent either fails to locate the element that it tries to click on and keeps clicking on the nearby region, or fails to start typing in the string when it is supposed to do so.

- *Failure to take reasonable attempts at all* refers to the failure mode where there is no clear reason that the agent fails to complete the task and does not seem to be on the right track throughout the trajectory.

- *Quit or press HOME early* refers to the failure mode where the agent decided to finish the task or press HOME to start over before the task is actually finished.

- *Stops at wrong but relevant page* refers to the failure mode where the agent arrives at a wrong page and mistakenly thinks that it had completed the task. For example, the agent finds a macbook on costco.com while the instruction asked it to find a macbook on ebay.com.

- *Technical issues* refer to the failure mode that either the task is impossible (e.g. the tasks asks to open Amazon app but this app is not installed) or the agent is temporarily blocked from a certain website due to frequent visits.

The translation between fine-grained failure modes and coarse-grained failure modes is presented in Table 5.

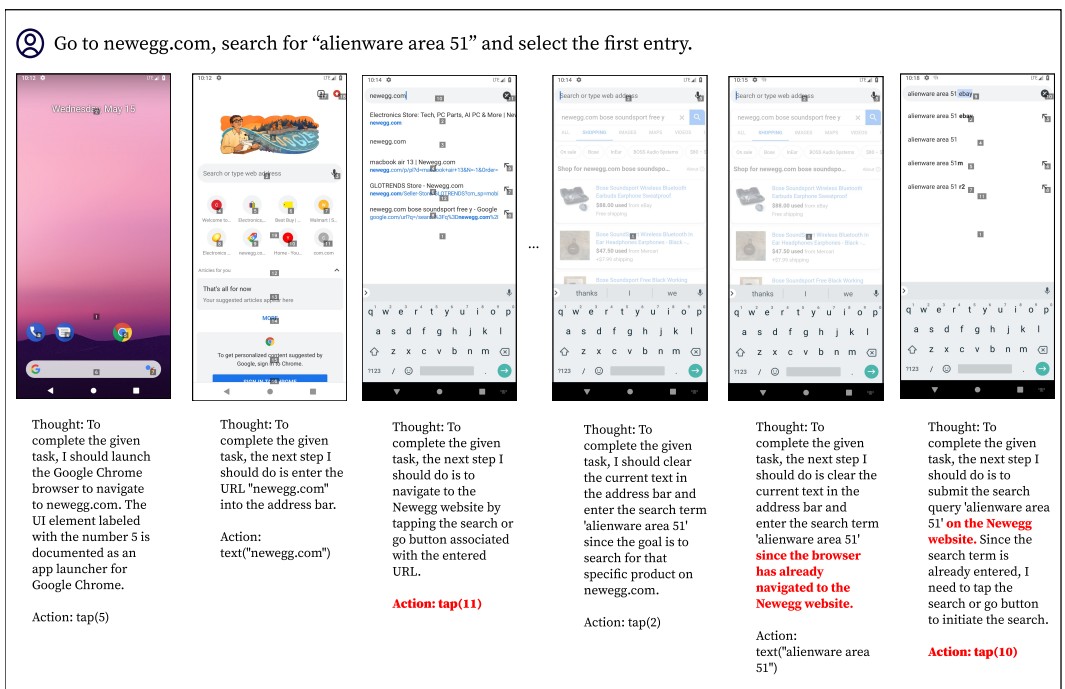

Figure 17: Failure of GPT-4V, with its thoughts and link-based actions given. A typical cause of failure is that it cannot tap on the correct "search" button after entering a query and mistakenly tapped onto the "x" symbol in the search bar as the "search" button. Here the goal is: Go to newegg.com, search for "alienware area 51" and select the first entry. As seen in red emboldened actions, it fails to press search button and deletes the query instead. Also, as seen in red highlighted parts in thoughts, it thinks it is in "newegg.com" website even though it is not.

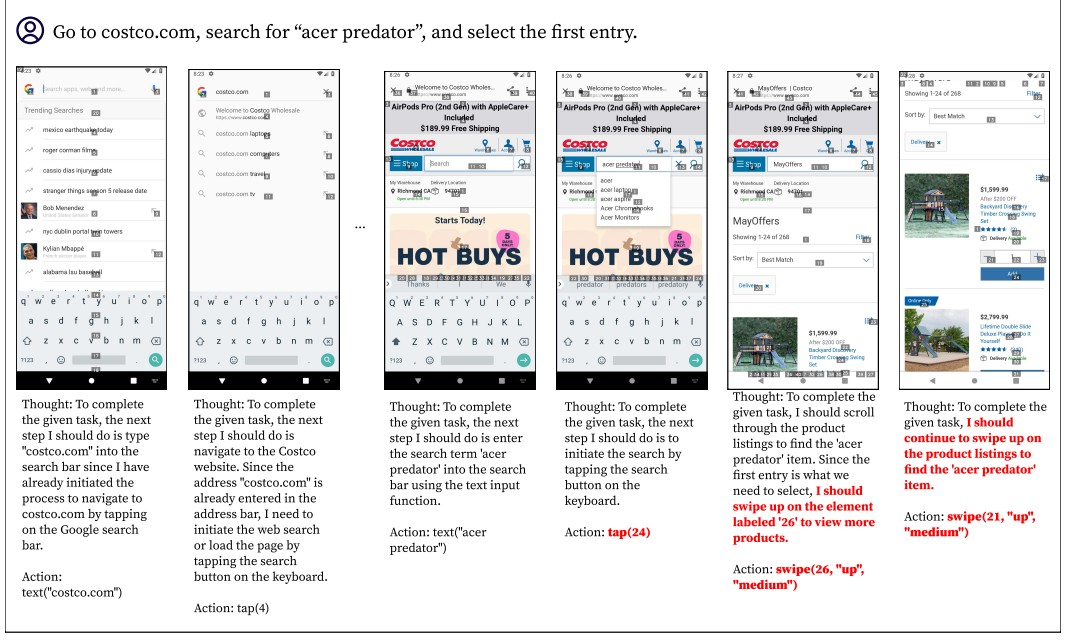

Figure 18: Failure of GPT-4V, with its thoughts and link-based actions given. This time the reason for failure is misclick on the wrong button. The task is "Go to costco.com, search for "acer predator", and select the first entry". Notice that up until the fourth frame in this Figure, the trajectory goes correct. But then it clicks on the generic advertisements on the Costco.com website, and it cannot recover back. It continues to scroll the page and takes wrong actions thereafter.

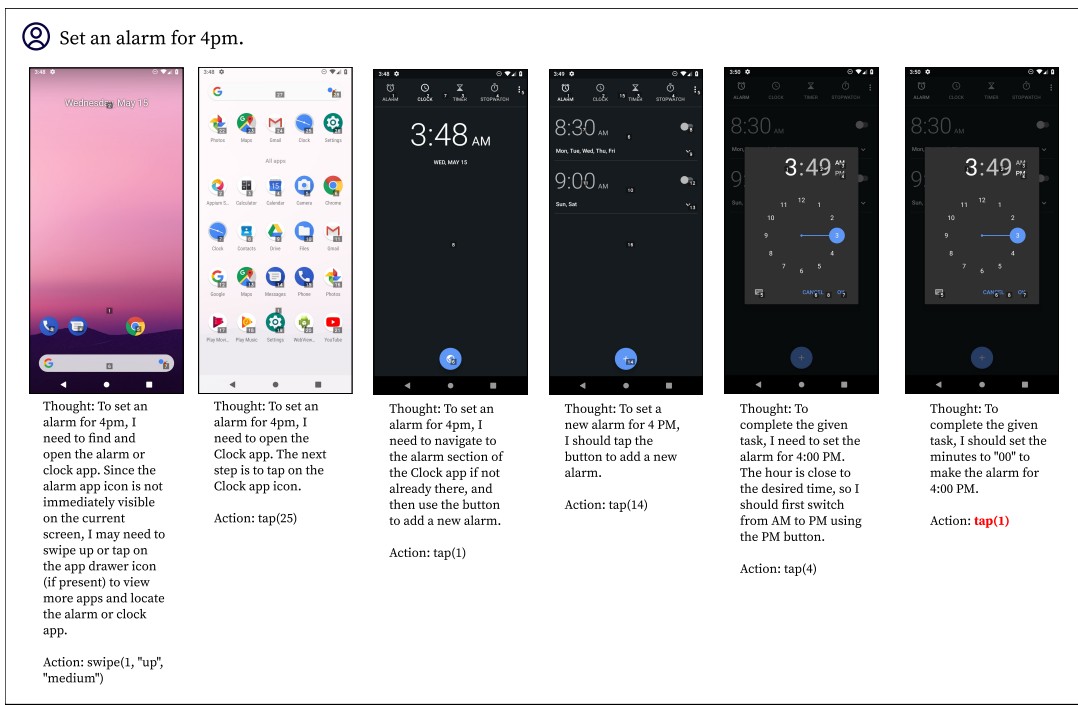

Figure 19: Failure of GPT-4V, with an example task on the AiTW general test set. The task is "Set an alarm for 4pm". Here, GPT-4V is able to successfully navigate to the clock app, and the alarm settings of that app. However, it cannot take the correct precise actions to set the alarm quickly enough, and it fails due to maximum rounds reached. In the last round, notice that the action of tap(1) contradict with its own thought process of setting minutes to "00".

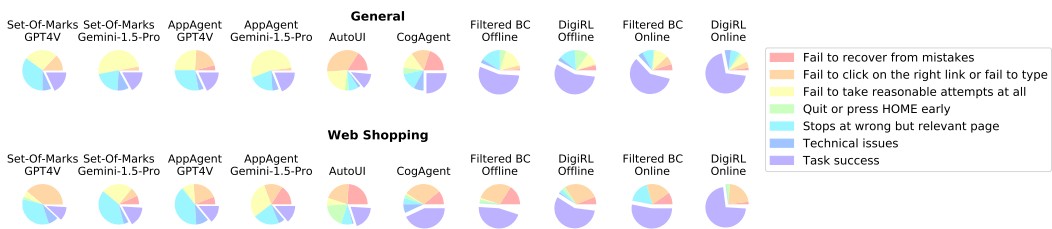

Figure 20: Failure modes decomposition for each policy model for both General and Web Shopping subsets.

| Fine-Grained Failure | Coarse-Grained Failure |
|---|---|
| Fail to recover from mistakes | Fail to recover from mistakes |
| Fail to click on the right link or fail to type | Get stuck midway |
| Fail to take reasonable attempts at all | Get stuck midway |
| Quit or Press HOME early | Arrive at wrong goal |
| Stops at wrong but relevant page | Arrive at wrong goal |
| Technical Issues | None |

Table 5: Examples of task descriptions in the AiTW Webshopping task set.

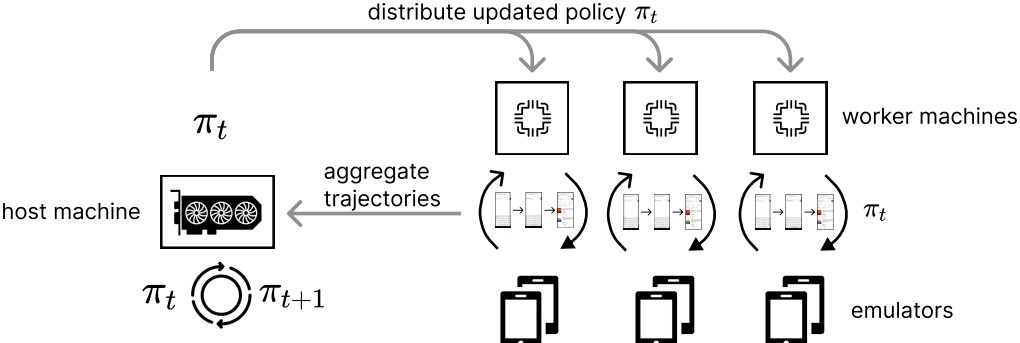

Figure 21: Multi-machine parallel emulator execution. The host machine is equipped with GPU accelerators and the worker machines are equipped only with CPUs. The policy update is executed on the worker machine and the trajectory collections are executed distributedly on the worker machines and aggregated by the host machine.

# E    Experiment machines

Our main experiments are conducted on VM instances from Google Cloud Platform. Each VM instance comes with 1x Tesla T4 GPU and 16x Intel(R) Xeon(R) CPU.

# F    Setup for parallel environment

Running multiple emulators in parallel can be challenging due to the inefficiency in thread synchronization and frequent fault propagation when one emulator runs into an unknown error. To address this challenge, we set up a server-client system where all emulator processes are running in independent server processes. Each emulator process communicates with the main training process through different UIAutomotor servers. The main training process sends high-level instructions to UIAutomotor servers (such as reset and step), while UIAutomotor servers parse high-level instructions into low-level UI commands (such as typing a character and tapping at a coordinate) and such UI commands are executed by the emulator processes. When an exception is thrown in the emulator, the UIAutomotor examines if it is recoverable (e.g. an UI command takes too long to execute in the emulator) and reset the emulator process if it is not. When an exception is thrown in the UIAutomotor server, the main training process stops and resets the UIAutomotor server to ensure data correctness.

This design can easily be scaled up to a multi-machine setting. As illustrated in Figure 21, one host machine equipped with GPU accelerator has a local copy of the current policy $\pi_t$, and distributes the policy to all worker machines equipped with only one GPU and multiple CPUs. Each worker machine will then collect trajectories of different tasks using $\pi_t$. After all collection processes are synchronized, the host machine gathers all the trajectories together to update the policy to $\pi_{t+1}$. This process keeps iterating until the policy converges.

**Speedup of emulation parallel.** The performance boost with respect to the number of worker machines is nearly linear, as demonstrated in Figure 22 (right), where we conduct experiments that examine the scaling performance of our parallel emulator. Our distributed emulator that runs emulations across multiple servers can reliably collect data with up to 64 parallel emulators on 128 CPUs with near-linear speedup. In contrast, a naive baseline that runs all parallel emulations on the same server achieves much inferior performance (0.74 compared to 1.74 trajs/min using 64 CPUs).

# G    Autonomous evaluator details

Our autonomous evaluator gives a reward to each observation we get. The observation is composed of the current screenshot of device and the task. The evaluator gives a reward of 1 if the screenshot shows a completion of the task, and will terminate the POMDP as a result result.

The optimized prompt is shown in Figure 23 and  Figure 24 for General and Web Shopping subsets respectively.

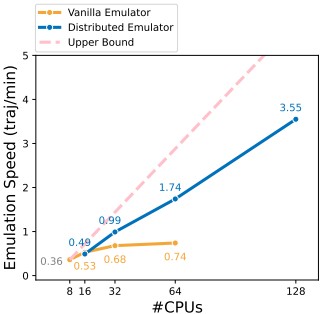

Figure 22: **Emulation speed w.r.t number of CPUs used.** The upper bound can only achieved when there is no communication and error handling cost. Our design of distributed emulator can significantly improve the efficiency of emulation compaared to the vanilla method of running all emulations over the same instance.

# H   Zero-shot Baseline Details

Figure 25 shows the prompt that we used for testing the Set-of-Marks performance for GPT-4V and Gemini 1.5 Pro. This prompt is directly taken from Yang et al. [47].

# I   Hyperparameters

Hyperparameters for both Filtered BC and DigiRL are carefully tuned through binary search on the training set of General and Web Shopping subsets. The final choice of hyperparameters for both methods can be found in Table 6. As shown in the table, the only hyperparameters introduced by DigiRL are supervised training hyperparameters for the value function and instruction value function (including number of iterations and learning rate) and GAE $\lambda$.

**Prompt**
You're an expert in evaluating whether the Screenshot successfully completes the Task.

**=====Examples=====**
Screenshot: {train_1.png}
Task: Open the settings.
Q: What should I expect to see on the screenshot if I've opened the settings?
A: I should expect to see I'm in the settings app. The screenshot shows the home screen of a mobile device, with various app icons displayed, including the settings app icon, but the settings app is not opened.
Status: failure

Screenshot: {train_2.png}
Task: Find hotels in washington dc
Q: What should I expect to see on the screenshot if I've searched for hotels in Washington, DC?
A: I should expect to see I'm in a search results page for hotels in Washington, DC. The screenshot shows a Google search page with the search field populated with the query "hotels in washington dc" and a list of suggested searches related to hotels in Washington, DC, but it does not show any search results for hotels in Washington, DC.
Status: failure

Screenshot: {train_3.png}
Task: What's a good restaurant in Portland?
Q: What should I expect to see on the screenshot if I've searched for a good restaurant in Portland?
A: I should expect to see I'm in a search results page for a good restaurant in Portland. The screenshot shows a Google search page with a search input field for "good restaurant in portland" and a map results preview showing business locations near Portland, like "Li Pigeon", "Portland City Grill", and "Higgins",
Status: success

... (more cases)

**=====Your Turn=====**
Screenshot: {test.png}
Task: {task_this_traj}
Respond in this format:
Q: What should I expect to see on the screenshot if I've <repeat the task>?
A: I should expect to see <first expectation, then what's in the given screenshot.>
Status: success or failure (don't return anything else)
Start with "Q:".

**Response**
Q: What should I expect to see on the screenshot if I've searched for the price of a 12' ladder at Home Depot?
A: I should expect to see the price of a 12' ladder at Home Depot; the screenshot shows a search result page for the price of a 12' ladder, with some product advertisements showing prices from Home Depot.
Status: success

**Image Sources**

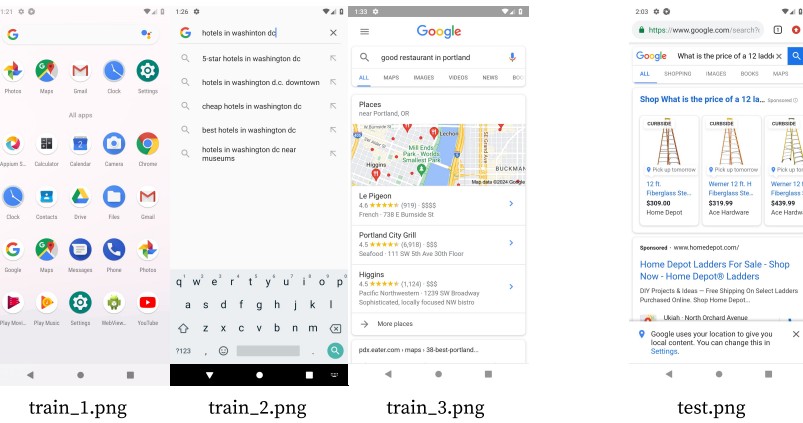

train_1.png          train_2.png          train_3.png          test.png

Figure 23: Prompt for our autonomous evaluator for tasks in AitW General subset.

**Prompt**

You're an expert in evaluating whether the Screenshot successfully completes the Task.

=====Examples=====
Screenshot: {train_1.png}
Task: Go to bestbuy.com
Q: What should I expect to see on the screenshot if I've gone to bestbuy.com?
A: I should expect to see I'm in the Best Buy website, which usually shows the best buy logo with some featured products and categories. The screenshot shows I'm searching for "bestbuy.com" in the Google search (with some search suggestions) instead of being in the Best Buy website.
Status: failure

Screenshot: {train_2.png}
Task: Go to ebay.com, search for "corsair k70"
Q: What should I expect to see on the screenshot if I've gone to ebay.com AND searched for "corsair k70"?
A: I should expect to see I'm in the eBay website and search results for "corsair k70". The screenshot shows I'm in the eBay ebay website with some search suggestions for "corsair k70", but it does not show search results of the product, which usually includes price and the product details.
Status: failure

Screenshot: {train_3.png}
Task: Go to ebay.com, search for "lenovo thinkpad"
Q: What should I expect to see on the screenshot if I've gone to ebay.com AND searched for "lenovo thinkpad"?
A: I should expect to see I'm in the eBay website and search results for "lenovo thinkpad". The screenshot shows I'm in the eBay website and have several search results for "lenovo thinkpad".
Status: success

... (more cases)

=====Your Turn=====
Screenshot: {test.png}
Task: {task_this_traj}
Respond in this format:
Q: What should I expect to see on the screenshot if I've <repeat the task>?
A: I should expect to see <first expectation, then what's in the given screenshot.>
Status: success or failure (don't return anything else)
Start with "Q:".

**Response**

Q: What should I expect to see on the screenshot if I've searched for the price of a 12' ladder at Home Depot?
A: I should expect to see the price of a 12' ladder at Home Depot; the screenshot shows a search result page for the price of a 12' ladder, with some product advertisements showing prices from Home Depot.
Status: success

**Image Sources**

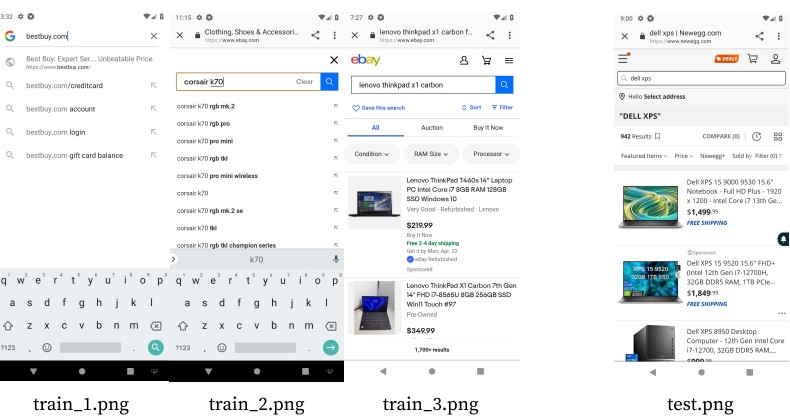

train_1.png      train_2.png      train_3.png      test.png

Figure 24: Prompt for our autonomous evaluator for tasks in AitW Web Shopping subset.

**Prompt**

"You are an agent that is trained to perform some basic tasks on a smartphone. You will be given a \nsmartphone screenshot. The interactive UI elements on the screenshot are labeled with numeric tags starting from 1. The \nnumeric tag of each interactive element is located in the center of the element.\n\nYou can call the following functions to control the smartphone:\n\n1. tap(element: int)\nThis function is used to tap an UI element shown on the smartphone screen.\n\"element\" is a numeric tag assigned to an UI element shown on the smartphone screen. \nA simple use case can be tap(5), which taps the UI element labeled with the number 5.\n\n2. text(text_input: str)\nThis function is used to insert text input in an input field/box. text_input is the string you want to insert and must \nbe wrapped with double quotation marks. A simple use case can be text(\"Hello, world!\"), which inserts the string \n\"Hello, world!\" into the input area on the smartphone screen. This function is usually callable when you see a keyboard \nshowing in the lower half of the screen.\n\n3. long_press(element: int)\nThis function is used to long press an UI element shown on the smartphone screen.\n\"element\" is a numeric tag assigned to an UI element shown on the smartphone screen.\nA simple use case can be long_press(5), which long presses the UI element labeled with the number 5.\n\n4. swipe(element: int, direction: str, dist: str)\nThis function is used to swipe an UI element shown on the smartphone screen, usually a scroll view or a slide bar.\n\"element\" is a numeric tag assigned to an UI element shown on the smartphone screen. \"direction\" is a string that \nrepresents one of the four directions: up, down, left, right. \"direction\" must be wrapped with double quotation \nmarks. \"dist\" determines the distance of the swipe and can be one of the three options: short, medium, long. You should \nchoose the appropriate distance option according to your need.\nA simple use case can be swipe(21, \"up\", \"medium\"), which swipes up the UI element labeled with the number 21 for a \nmedium distance.\n\n5. grid()\nYou should call this function when you find the element you want to interact with is not labeled with a numeric tag and \nother elements with numeric tags cannot help with the task. The function will bring up a grid overlay to divide the \nsmartphone screen into small areas and this will give you more freedom to choose any part of the screen to tap, long \npress, or swipe.

The task you need to complete is to **How much does a 2 bedroom apartment rent for in Denver?**.

Your past actions to proceed with this task are summarized as follows: **None**

Now, given the documentation and the following labeled screenshot, you need to think and call the function needed to proceed with the task. Your output should include three parts in the given format:
Observation: <Describe what you observe in the image>
Thought: <To complete the given task, what is the next step I should do>
Action: <The function call with the correct parameters to proceed with the task. When you are certain that the task is successfully done and the goal is reached as of the current observation, you should output FINISH. You cannot output anything else except a function call or FINISH \nin this field.>
Summary: <Summarize your past actions along with your latest action in one or two sentences. Do not include the numeric \ntag in your summary>\nYou can only take one action at a time, so please directly call the function."

Figure 25: Set-of-Marks prompting. The boldened inputs can be changed according to our goal. The task changes for every different task. The past actions change as we take actions (it is None now since this is the prompt for the first round).


Table 6: Hyperparameters for All Experiments

| Method | Hyperparameter | Offline | Offline-to-Online |
|---|---|---|---|
| Filtered BC | actor lr | 3e-3 | 3e-3 |
| | batch size | 128 | 128 |
| | rollout trajectories | - | 16 |
| | replay buffer size | - | 5000 |
| | rollout temperature | - | 1.0 |
| | maximum gradient norm | 0.01 | 0.01 |
| | actor updates per iteration | 20 | 20 |
| | number of iterations for offline actor updates | 10 | 10 |
| DigiRL | actor lr | 3e-3 | 3e-3 |
| | value function lr | 3e-3 | 3e-3 |
| | instruction value function lr | 3e-3 | 3e-3 |
| | instruction value function lr | 3e-3 | 3e-3 |
| | batch size | 128 | 128 |
| | rollout trajectories | - | 16 |
| | replay buffer size | - | 5000 |
| | rollout temperature | - | 1.0 |
| | maximum gradient norm | 0.01 | 0.01 |
| | GAE $\lambda$ | 0.5 | 0.5 |
| | actor updates per iteration | 20 | 20 |
| | value function updates per iteration | 5 | 5 |
| | instruction value function updates per iteration | - | 5 |
| | number of iterations for offline actor updates | 10 | 10 |
| | number of iterations for offline value function updates | 20 | 20 |
| | number of iterations for offline instruction value function updates | - | 20 |

Table 7: Hyperparameters for DigiRL and Filtered BC on both General and Web Shopping subset of AitW..

- It is fine to include aspirational goals as motivation as long as it is clear that these goals are not attained by the paper.

2. **Limitations**

Question: Does the paper discuss the limitations of the work performed by the authors?

Answer: [Yes]

Justification: Limitations are discussed in the last section of the paper.

Guidelines:

- The answer NA means that the paper has no limitation while the answer No means that the paper has limitations, but those are not discussed in the paper.
- The authors are encouraged to create a separate "Limitations" section in their paper.
- The paper should point out any strong assumptions and how robust the results are to violations of these assumptions (e.g., independence assumptions, noiseless settings, model well-specification, asymptotic approximations only holding locally). The authors should reflect on how these assumptions might be violated in practice and what the implications would be.
- The authors should reflect on the scope of the claims made, e.g., if the approach was only tested on a few datasets or with a few runs. In general, empirical results often depend on implicit assumptions, which should be articulated.
- The authors should reflect on the factors that influence the performance of the approach. For example, a facial recognition algorithm may perform poorly when image resolution is low or images are taken in low lighting. Or a speech-to-text system might not be

used reliably to provide closed captions for online lectures because it fails to handle technical jargon.

- The authors should discuss the computational efficiency of the proposed algorithms and how they scale with dataset size.
- If applicable, the authors should discuss possible limitations of their approach to address problems of privacy and fairness.
- While the authors might fear that complete honesty about limitations might be used by reviewers as grounds for rejection, a worse outcome might be that reviewers discover limitations that aren't acknowledged in the paper. The authors should use their best judgment and recognize that individual actions in favor of transparency play an important role in developing norms that preserve the integrity of the community. Reviewers will be specifically instructed to not penalize honesty concerning limitations.

3. **Theory Assumptions and Proofs**

   Question: For each theoretical result, does the paper provide the full set of assumptions and a complete (and correct) proof?

   Answer: [NA]

   Justification: This paper does not provide theoretical results.

   Guidelines:

   - The answer NA means that the paper does not include theoretical results.
   - All the theorems, formulas, and proofs in the paper should be numbered and cross-referenced.
   - All assumptions should be clearly stated or referenced in the statement of any theorems.
   - The proofs can either appear in the main paper or the supplemental material, but if they appear in the supplemental material, the authors are encouraged to provide a short proof sketch to provide intuition.
   - Inversely, any informal proof provided in the core of the paper should be complemented by formal proofs provided in appendix or supplemental material.
   - Theorems and Lemmas that the proof relies upon should be properly referenced.

4. **Experimental Result Reproducibility**

   Question: Does the paper fully disclose all the information needed to reproduce the main experimental results of the paper to the extent that it affects the main claims and/or conclusions of the paper (regardless of whether the code and data are provided or not)?

   Answer: [Yes]

   Justification: All loss functions and implementation details are provided in Section 4.

   Guidelines:

   - The answer NA means that the paper does not include experiments.
   - If the paper includes experiments, a No answer to this question will not be perceived well by the reviewers: Making the paper reproducible is important, regardless of whether the code and data are provided or not.
   - If the contribution is a dataset and/or model, the authors should describe the steps taken to make their results reproducible or verifiable.
   - Depending on the contribution, reproducibility can be accomplished in various ways. For example, if the contribution is a novel architecture, describing the architecture fully might suffice, or if the contribution is a specific model and empirical evaluation, it may be necessary to either make it possible for others to replicate the model with the same dataset, or provide access to the model. In general. releasing code and data is often one good way to accomplish this, but reproducibility can also be provided via detailed instructions for how to replicate the results, access to a hosted model (e.g., in the case of a large language model), releasing of a model checkpoint, or other means that are appropriate to the research performed.
   - While NeurIPS does not require releasing code, the conference does require all submissions to provide some reasonable avenue for reproducibility, which may depend on the nature of the contribution. For example

(a) If the contribution is primarily a new algorithm, the paper should make it clear how to reproduce that algorithm.

(b) If the contribution is primarily a new model architecture, the paper should describe the architecture clearly and fully.

(c) If the contribution is a new model (e.g., a large language model), then there should either be a way to access this model for reproducing the results or a way to reproduce the model (e.g., with an open-source dataset or instructions for how to construct the dataset).

(d) We recognize that reproducibility may be tricky in some cases, in which case authors are welcome to describe the particular way they provide for reproducibility. In the case of closed-source models, it may be that access to the model is limited in some way (e.g., to registered users), but it should be possible for other researchers to have some path to reproducing or verifying the results.

5. **Open access to data and code**

Question: Does the paper provide open access to the data and code, with sufficient instructions to faithfully reproduce the main experimental results, as described in supplemental material?

Answer: [No]

Justification: We are still actively cleaning the code and make the environment more accessible to a broader audience. Once we are done with that, we will open-source the code along with the release of the paper.

Guidelines:

- The answer NA means that paper does not include experiments requiring code.
- Please see the NeurIPS code and data submission guidelines (https://nips.cc/public/guides/CodeSubmissionPolicy) for more details.
- While we encourage the release of code and data, we understand that this might not be possible, so "No" is an acceptable answer. Papers cannot be rejected simply for not including code, unless this is central to the contribution (e.g., for a new open-source benchmark).
- The instructions should contain the exact command and environment needed to run to reproduce the results. See the NeurIPS code and data submission guidelines (https://nips.cc/public/guides/CodeSubmissionPolicy) for more details.
- The authors should provide instructions on data access and preparation, including how to access the raw data, preprocessed data, intermediate data, and generated data, etc.
- The authors should provide scripts to reproduce all experimental results for the new proposed method and baselines. If only a subset of experiments are reproducible, they should state which ones are omitted from the script and why.
- At submission time, to preserve anonymity, the authors should release anonymized versions (if applicable).
- Providing as much information as possible in supplemental material (appended to the paper) is recommended, but including URLs to data and code is permitted.

6. **Experimental Setting/Details**

Question: Does the paper specify all the training and test details (e.g., data splits, hyperparameters, how they were chosen, type of optimizer, etc.) necessary to understand the results?

Answer: [Yes]

Justification: Dataset details are provided in Appendix A.1 and hyperparameters are provided in Appendix I.

Guidelines:

- The answer NA means that the paper does not include experiments.
- The experimental setting should be presented in the core of the paper to a level of detail that is necessary to appreciate the results and make sense of them.
- The full details can be provided either with the code, in appendix, or as supplemental material.

7. **Experiment Statistical Significance**

   Question: Does the paper report error bars suitably and correctly defined or other appropriate information about the statistical significance of the experiments?

   Answer: [Yes]

   Justification: Repeated experiments are carried out with their means and standard deviations reported in Table 1.

   Guidelines:

   - The answer NA means that the paper does not include experiments.
   - The authors should answer "Yes" if the results are accompanied by error bars, confidence intervals, or statistical significance tests, at least for the experiments that support the main claims of the paper.
   - The factors of variability that the error bars are capturing should be clearly stated (for example, train/test split, initialization, random drawing of some parameter, or overall run with given experimental conditions).
   - The method for calculating the error bars should be explained (closed form formula, call to a library function, bootstrap, etc.)
   - The assumptions made should be given (e.g., Normally distributed errors).
   - It should be clear whether the error bar is the standard deviation or the standard error of the mean.
   - It is OK to report 1-sigma error bars, but one should state it. The authors should preferably report a 2-sigma error bar than state that they have a 96% CI, if the hypothesis of Normality of errors is not verified.
   - For asymmetric distributions, the authors should be careful not to show in tables or figures symmetric error bars that would yield results that are out of range (e.g. negative error rates).
   - If error bars are reported in tables or plots, The authors should explain in the text how they were calculated and reference the corresponding figures or tables in the text.

8. **Experiments Compute Resources**

   Question: For each experiment, does the paper provide sufficient information on the computer resources (type of compute workers, memory, time of execution) needed to reproduce the experiments?

   Answer: [Yes]

   Justification: This information is provided in Appendix E.

   Guidelines:

   - The answer NA means that the paper does not include experiments.
   - The paper should indicate the type of compute workers CPU or GPU, internal cluster, or cloud provider, including relevant memory and storage.
   - The paper should provide the amount of compute required for each of the individual experimental runs as well as estimate the total compute.
   - The paper should disclose whether the full research project required more compute than the experiments reported in the paper (e.g., preliminary or failed experiments that didn't make it into the paper).

9. **Code Of Ethics**

   Question: Does the research conducted in the paper conform, in every respect, with the NeurIPS Code of Ethics https://neurips.cc/public/EthicsGuidelines?

   Answer: [Yes]

   Justification: The research conducted in the paper conform, in every respect, with the NeuIPS code of Etics.

   Guidelines:

   - The answer NA means that the authors have not reviewed the NeurIPS Code of Ethics.

- If the authors answer No, they should explain the special circumstances that require a deviation from the Code of Ethics.
- The authors should make sure to preserve anonymity (e.g., if there is a special consideration due to laws or regulations in their jurisdiction).

10. **Broader Impacts**

Question: Does the paper discuss both potential positive societal impacts and negative societal impacts of the work performed?

Answer: [Yes]

Justification: The positive societal impacts are discussed in the Introduction while the negative societal impacts are discussed in Section 6.

Guidelines:

- The answer NA means that there is no societal impact of the work performed.
- If the authors answer NA or No, they should explain why their work has no societal impact or why the paper does not address societal impact.
- Examples of negative societal impacts include potential malicious or unintended uses (e.g., disinformation, generating fake profiles, surveillance), fairness considerations (e.g., deployment of technologies that could make decisions that unfairly impact specific groups), privacy considerations, and security considerations.
- The conference expects that many papers will be foundational research and not tied to particular applications, let alone deployments. However, if there is a direct path to any negative applications, the authors should point it out. For example, it is legitimate to point out that an improvement in the quality of generative models could be used to generate deepfakes for disinformation. On the other hand, it is not needed to point out that a generic algorithm for optimizing neural networks could enable people to train models that generate Deepfakes faster.
- The authors should consider possible harms that could arise when the technology is being used as intended and functioning correctly, harms that could arise when the technology is being used as intended but gives incorrect results, and harms following from (intentional or unintentional) misuse of the technology.
- If there are negative societal impacts, the authors could also discuss possible mitigation strategies (e.g., gated release of models, providing defenses in addition to attacks, mechanisms for monitoring misuse, mechanisms to monitor how a system learns from feedback over time, improving the efficiency and accessibility of ML).

11. **Safeguards**

Question: Does the paper describe safeguards that have been put in place for responsible release of data or models that have a high risk for misuse (e.g., pretrained language models, image generators, or scraped datasets)?

Answer: [NA]

Justification: The capability of the model that we will be releasing is limited to simple tasks in Android in the Wild dataset, and therefore does not have a high risk for misuse.

Guidelines:

- The answer NA means that the paper poses no such risks.
- Released models that have a high risk for misuse or dual-use should be released with necessary safeguards to allow for controlled use of the model, for example by requiring that users adhere to usage guidelines or restrictions to access the model or implementing safety filters.
- Datasets that have been scraped from the Internet could pose safety risks. The authors should describe how they avoided releasing unsafe images.
- We recognize that providing effective safeguards is challenging, and many papers do not require this, but we encourage authors to take this into account and make a best faith effort.

12. **Licenses for existing assets**

Question: Are the creators or original owners of assets (e.g., code, data, models), used in the paper, properly credited and are the license and terms of use explicitly mentioned and properly respected?

Answer: [Yes]

Justification: We have properly cited the assets that we are using.

Guidelines:

- The answer NA means that the paper does not use existing assets.
- The authors should cite the original paper that produced the code package or dataset.
- The authors should state which version of the asset is used and, if possible, include a URL.
- The name of the license (e.g., CC-BY 4.0) should be included for each asset.
- For scraped data from a particular source (e.g., website), the copyright and terms of service of that source should be provided.
- If assets are released, the license, copyright information, and terms of use in the package should be provided. For popular datasets, paperswithcode.com/datasets has curated licenses for some datasets. Their licensing guide can help determine the license of a dataset.
- For existing datasets that are re-packaged, both the original license and the license of the derived asset (if it has changed) should be provided.
- If this information is not available online, the authors are encouraged to reach out to the asset's creators.

13. **New Assets**

Question: Are new assets introduced in the paper well documented and is the documentation provided alongside the assets?

Answer: [NA]

Justification: This submission does not include new assets. New assets including opensourced code, model checkpoints, and model trajectories will be released with documentation when we release the paper.

Guidelines:

- The answer NA means that the paper does not release new assets.
- Researchers should communicate the details of the dataset/code/model as part of their submissions via structured templates. This includes details about training, license, limitations, etc.
- The paper should discuss whether and how consent was obtained from people whose asset is used.
- At submission time, remember to anonymize your assets (if applicable). You can either create an anonymized URL or include an anonymized zip file.

14. **Crowdsourcing and Research with Human Subjects**

Question: For crowdsourcing experiments and research with human subjects, does the paper include the full text of instructions given to participants and screenshots, if applicable, as well as details about compensation (if any)?

Answer: [NA]

Justification: This research does not involve crowdsourcing or human subjects. Annotations of trajectories in Figure 7 and Figure 8 are carried out by authors alone.

Guidelines:

- The answer NA means that the paper does not involve crowdsourcing nor research with human subjects.
- Including this information in the supplemental material is fine, but if the main contribution of the paper involves human subjects, then as much detail as possible should be included in the main paper.

- According to the NeurIPS Code of Ethics, workers involved in data collection, curation, or other labor should be paid at least the minimum wage in the country of the data collector.

15. **Institutional Review Board (IRB) Approvals or Equivalent for Research with Human Subjects**

    Question: Does the paper describe potential risks incurred by study participants, whether such risks were disclosed to the subjects, and whether Institutional Review Board (IRB) approvals (or an equivalent approval/review based on the requirements of your country or institution) were obtained?

    Answer: [NA]

    Justification: This research does not involve crowdsourcing or human subjects.

    Guidelines:
    - The answer NA means that the paper does not involve crowdsourcing nor research with human subjects.
    - Depending on the country in which research is conducted, IRB approval (or equivalent) may be required for any human subjects research. If you obtained IRB approval, you should clearly state this in the paper.
    - We recognize that the procedures for this may vary significantly between institutions and locations, and we expect authors to adhere to the NeurIPS Code of Ethics and the guidelines for their institution.
    - For initial submissions, do not include any information that would break anonymity (if applicable), such as the institution conducting the review.

