# OpenReview forum: "DigiRL: Training In-The-Wild Device-Control Agents with Autonomous Reinforcement Learning"
_NeurIPS.cc/2024/Conference — NeurIPS 2024 poster_

### Official Review · Reviewer_zDzF · 2024-07-13

**Soundness:** 3
**Presentation:** 3
**Contribution:** 3
**Rating:** 7
**Confidence:** 3

**Summary:**

This paper shows the potential of Reinforcement Learning (RL) for designing an effective digital agent for in-the-wild control through Graphical User Interfaces (GUIs). The proposed approach relies on the advantage of the pre-trained visual language models (VLMs) while tackling real-world stochasticity by training an RL agent that interacts with an environment instead of relying on static demonstrations. Accordingly, this work proposes a novel autonomous RL approach, namely DigiRL, for training device control agents, which consists of two stages: an offline RL phase where the agent is trained on static demonstrations, followed by an in-the-wild, offline-to-online RL stage for training the agent through interacting with an environment. Consequently, this work also introduces a scalable and parallelizable Android learning environment with a reward model (evaluator) based on a robust VLM-based model. To show the effectiveness of the proposed method, an evaluation of different tasks given diverse instructions is carried out from the Android in the Wild dataset on real Android device emulators. The results show a significant improvement of DigiRL compared to the existing state-of-the-art agents, including wrapped proprietary VLMs such as GPT-4V and Gemini 1.5 pro. The paper claims to be the first to succeed in developing an autonomous offline-to-online RL approach to enable state-of-the-art performance on device control problems.

**Strengths:**

- The usage of RL in designing a successful digital agent for a device control task is fascinating.
- I appreciate implementing such a scalable Android learning environment, and I hope the authors will open-source everything so that other researchers can reuse it.
- The clarity of the paper is worth mentioning.
- The experimental results section is rich, especially the ablation studies.

**Weaknesses:**

- Although the POMDP definition sounds correct when defining the problem, a contextual POMDP is even more appropriate for such a problem [1].
- A pseudo-code or an illustrative diagram should be added to facilitate understanding the method.
- It is not clear how the policy and the value network are conditioned given the context $c$. (implementation-wise)

[1] Hallak, Assaf, Dotan Di Castro, and Shie Mannor. "Contextual markov decision processes." arXiv preprint arXiv:1502.02259 (2015).

**Questions:**

- How are the policy and the value network conditioned given the context $c$?
- Is the RL agent trained in a multi-task fashion (which I believe is true)? I mean the agent is trained with more than one task concurrently accessing fully or partially the same models.
- If yes, do you think classical Multi-task learning or Multi-task reinforcement learning approaches would enhance the performance even more?

**Limitations:**

I believe the authors discussed the method's limitations in the final section. I agree with the authors regarding the impact of such application on the economy, society, and privacy, and that needs careful review in the future to limit any harm.

---

> ### Author Rebuttal · Authors · 2024-08-07
>
> Thank you for your review and feedback on the paper. We provide responses to the questions raised below that we will also include in the updated version of the paper. We commit to open sourcing open-source the code, environment, checkpoint, and the data. In this rebuttal period, we provide an anonymous link to our code: (link sent via a private message to AC per rebuttal instruction), and will make this public  with the final version.
>
> Thanks so much for your appreciation of our work that “the usage of RL in designing a successful digital agent for device control tasks is fascinating”! **Please let us know if your concerns are addressed and if so, we would appreciate it if you might be willing to upgrade your score.**  We answer your questions below:
>
> ___
>
> ### “It is not clear how the policy and the value network are conditioned given the context c (implementation-wise)”
>
> Implementation-wise, we use a 1B VLM AutoUI for the policy network so the context (in our case the task to complete) is directly included in the language input / encoder module of the VLM. For the value network, we encode the image with a CLIP encoder and the context with 110M BERT-base encoder for computational efficiency reasons. After this, an additional MLP layer is added on top of CLIP and BERT encodings concatenated with each other. We will include this clarification in the implementation detail section of a revised version of the paper.
>
> ___
>
> ### “A pseudo-code or an illustrative diagram should be added to facilitate understanding the method.”
>
> Thanks for the suggestion. We have made an illustrative diagram to facilitate readers to understand the method in Figure 2 of Rebuttal PDF. In this diagram, we have an instruction-level value function and a step-level value function, both of which serve as “filters” to identify the “advantageous” data that the agent should try to train on with AWR. We will include this diagram in an updated version of the paper!
>
> ___
>
> ### “Is the RL agent trained in a multi-task fashion (which I believe is true)? I mean the agent is trained with more than one task concurrently accessing fully or partially the same models.”
>
>
> Yes, the RL agent is trained in a multi-task fashion where at the start of each episode a random task (e.g. “Find the nearest place to buy a beach umbrella” and “Go to ebay.com, search for logitech 950” ) is drawn out of  a task pool containing 200 tasks.
>
> ___
>
> ### “If yes, do you think classical Multi-task learning or Multi-task reinforcement learning approaches would enhance the performance even more?”
> This is a good idea! In fact, we believe that DigiRL is an effective starting point for building effective device control agents and researching RL algorithms for training agents. We are now ourselves building on DigiRL, to devise better RL approaches for training agents in this environment. Your suggestion is valuable for our exploration and we will study multi-task RL approaches such as PC-Grad [1], Task Grouping [2], etc to further enhance performance of DigiRL.
>
> ___
>
> ### “Although the POMDP definition sounds correct when defining the problem, a contextual POMDP is even more appropriate for such a problem”
>
> Thanks for pointing out this definition! We will post the device control problem in Section 3 as a contextual POMDP as it is indeed more appropriate. Note that this does not change any of the training objectives as it is largely a notational change. We will include an additional distribution over contexts $c$ when we define the POMDP.
>
> [1] Yu, Tianhe, et al. ‘Gradient Surgery for Multi-Task Learning’. arXiv [Cs.LG], 2020, http://arxiv.org/abs/2001.06782. arXiv.
>
> [2] Fifty, Christopher, et al. ‘Efficiently Identifying Task Groupings for Multi-Task Learning’. arXiv [Cs.LG], 2021, http://arxiv.org/abs/2109.04617. arXiv.

---

> ### Author Response · Authors · 2024-08-07
> **Link to annonymous code**
>
> https://anonymous.4open.science/r/digirl-anonymous-7ED0/
> Here is the link to our code promised for Reviewer zDzF

---

> ### Comment · Reviewer_zDzF · 2024-08-13
> **Rebuttal**
>
> Dear Authors,
>
> Thanks a lot for answering my questions and addressing my concerns!
>
> Given the authors' responses to my questions and those of other reviewers, I will increase my score from 6->7 while keeping my confidence level, as it is similar to Reviewer 251X; my background is RL.

---

> > ### Author Response · Authors · 2024-08-13
> >
> > We thank the reviewer for the response. We appreciate your score increase!

---

### Official Review · Reviewer_251X · 2024-07-13

**Soundness:** 2
**Presentation:** 3
**Contribution:** 2
**Rating:** 5
**Confidence:** 3

**Summary:**

This paper introduces a novel autonomous reinforcement learning (RL) approach, DigiRL, for training in-the-wild device control agents. DigiRL first employs offline RL to fine-tune a pre-trained vision-language model (VLM as the agent) using stale task-specific data, and then further refines the agent through online RL by continuously interacting with parallelized emulators. DigiRL achieves a 49.5% absolute improvement in task success rate over existing state-of-the-art agents, establishing a new benchmark for digital agents in device control.

**Strengths:**

1. The paper is well-written and motivated, many important technical/implementation details are covered.

2. The paper considers a challenging problem, autonomous device control, where existing LLM-based methods struggle to achieve acceptable success rate. The proposed method leverages VLM and RL techniques and significantly improves compared to these baselines.

3. The experiments are comprehensive and informative, covering LLM/RL agents, prompting and learning paradigms, offline and off-to-on RL, as well as failure modes analysis.

4. The authors implement a multi-machine emulator system to support parallel and real-time training of online RL.

**Weaknesses:**

Major Points:

1. From a ML methodological point of view, the novelty/contribution of the paper is limited. To perform offline and off-to-on RL, the paper adopts a number of existing techniques such as AWR, doubly-robust estimators with little customization (e.g.,  hard filtering on the advantages instead of computing $\exp(A)$, which is mainly indended for easier implementation), all well-known to the community. The only thing seems "new" is training value functions with cross-entropy losses, also directly taken from [1], and the equations in line 250-251 seems to be qustionable (see my comments in the Questions section). Moreover, no theoretical insight is provided to elucidate why these specific designs are chosen.

[1] Stop regressing: Training value functions via classification for scalable deep rl, 2024.

2. Limited Scope. The entire paper focuses on a very specific domain (autonomous device control). The scope of the proposed method might be too narrow to be of general interest to the ML/RL community.

Minor Points:

- In section 4.2, how to properly balance the two estimators, one with higher variance and one with higher bias to achieve the optimal result? What's the hyperparameter profile of the combined estimator? Have you tried any alternative designs and can you give theoretical insight to justify this specific design choice?

- Regarding the offline and off-to-on RL setting: to my knowledge, the main point of offline RL is to leverage a large body of stale data to safely and efficiently pretrain a RL agent. Therefore, for off-to-on RL, where online RL operates as the fine-tuning stage, one should use data far less than the offline pretrained dataset to ensure the setting is meaningful. The fact that authors intentionally use the same amount of data for both offline and online stage, which assumes access to a large amount of online data might make the offline pretraining unecessary. To see this, I recommend the authors to directly perform online RL on the combined dataset and it's highly possible that such "purely online" agent outperforms its off-to-on counterpart.

-  The authors spend quite a few words discussing the challenges of stochasticity and device control as a POMDP. However, I do not see any specific design or techincal contribution targeting such problems.

**Questions:**

1. In line 250-251, the CE loss pairs r with log V and (1-r) with log (1-V). Intuitively, this means one would like to make the distribution of r and V as close as possible (when r-> 1, V -> 1 and vice versa). This seems to contradict the claim in line 240-241 that "Intuitively, if a rollout attains a high value of A(sh, ah, c), it means the value function V is small".

2. How do you perform the train/test task split, if not random split? It is odd to see in Table 1 that almost all testing performance clearly surpass the training performance (normally should be the opposite), which suggest that the testing tasks are in general easier than the training tasks and not i.i.d?

---

> ### Author Rebuttal · Authors · 2024-08-07
>
> Thank you for the review! At the outset, we want to clarify our scope: our goal is to show that autonomous RL can be used to build SOTA device control agents that outperform proprietary models (Gemini/GPT-4). Our methodological contribution involves identifying and designing a good RL objective to be able to do so (robust advantages + curriculum + cross-entropy loss + AWR). We believe that our scope of device control is akin to several prior ICML/ICLR/NeurIPS papers that focus on applying ML / RL techniques to one domain (e.g., RL for chip design (ICML) [6], LLMs for web navigation (ICLR)  [7], web shopping (NeurIPS) [1]), and were still judged to be valuable contributions. In fact, device control is already more general than several important agent domains (e.g., shopping, web navigation, traveling planning, etc) that have been considered individually. We therefore think that an RL approach that attains SoTA **for the first time**, in a more general setting than recent papers, should not be a ground for rejection. To address the questions, we add **new results for hyperparameter profiling, advantage design, online RL from scratch, and
> stochasticity**. *Please let us know if the concerns are addressed and if so, we would be grateful if you upgrade your score.*
> ___
>
> ### “Limited Scope; From a ML methodological point of view, the novelty/contribution of the paper is limited.”
>
> As mentioned, our work is already in a more general setting than work in foundation agents that appears in ICML / NeurIPS / ICLR [1,2,3,4]. We use two subsets of AitW that focus on different parts of device control with around 200 tasks each (web shopping, device management; see Tables 2, 3). For the RL community, we identify challenges in a real-world problem setting of device control (e.g., see Fig 1, Table 1 in the PDF), and design an RL approach that can efficiently learn in this environment by combining robust advantages, curriculum, cross-entropy losses & AWR. While each piece individually is not as novel, combining each piece into an effective system for a real-world, user-scale problem is our contribution. As a systems paper (that has been of interest in ML/RL [5,6,7]), we think we should be evaluated on the efficacy of our system rather than a novel RL algorithm.
> ___
>
> ### “Step level advantage estimator”
>
> **Alternate designs:** We tried using a Q-function $Q(s, a)$ for computing advantages, but this leads to significantly worse and less stable results (see Table 1 of the PDF). This is because the action coverage is not high enough to  properly attribute reward signals to the action instead of states.
>
> **The theoretical justification:** follows the analysis of GAE[8], where the one-step estimator $V^{step}(s_{h+1}) + r(s_h, a_h) - V^{step}(s_{h})$ corresponds to the high-bias estimator $\hat{A}^{(1)}$ in Eq. 11 of the GAE paper and the MC reward estimator $\lambda^{H-h}r(s_H, a_H, c)$ corresponds to the high-variance estimator $\hat{A}^{\infty}$ in Eq. 15 in GAE. While GAE takes an average of a series of k-step estimators, we choose to omit the intermediate estimators for simplicity, nonetheless we enjoy similar bias-variance trade-offs. Similar to the GAE[8], the combined estimator is $\gamma$-just (Proposition 1[8]) when $V^{step}$ is accurate, i.e.
> $E_{s,a \sim d^\pi}A^{step}(s,a)\nabla \pi_{\theta}(a|s) = E_{s,a \sim d^\pi}A^{\pi}(s,a)\nabla \pi_{\theta}(a|s)$, where $A^{\pi}$ is the true advantage function.
>
> **Hyperparam tuning:** The only hyperparameter is $\lambda$, which is tuned similar to the GAE $\lambda$. We provide a result in  **table 2 of 1-page PDF** ablating $\lambda$ from 0.0 to 0.9, and found DigiRL to not be very sensitive to it.
> ___
>
> ### Benefits of offline-to-online RL
>
> We ran online RL from scratch in Figure 3 of the PDF. We see that our off-to-on agent smoothly transitions into the online phase, without any unlearning + results in a lower cumulative regret than online RL from scratch. Avoiding unlearning while benefitting from a better init is crucial in scenarios where the desideratum is to keep adapting the agent quickly upon deployment.
> ___
>
> ### Handling stochasticity and POMDP via DigiRL
>
> To show the challenge of stochasticity and dynamism, we add Figure 1 in 1-page PDF comparing the performance of a stale offline DigiRL agent and an agent updated via online. Despite being trained with RL, performance of a stale agent decays as time moves on, whereas continually training with online DigiRL avoids this issue, hence DigiRL addresses performance drop amidst stochasticity.
>
> We also clarify that utilizing a POMDP formulation is important because at some instants, the exact state of the device may simply be unknown (e.g., when a page is loading it is impossible to know what is loading without referring to the previous state). Our practical implementation handles this by using the history of the last two screenshots as the state of the RL agent (see Lines 199-200).
> ___
>
> ### Line 250-251, clarification about r, V and cross-entropy
>
> This would not make r and V close because r is a function of both s & a, but V is only a function of the instruction. As long as the reward values for all (s,a) for a given instruction are not 1, then, $V$ will take a value smaller than 1. Concretely, $V$ is the average $r$ for all (s, a) pairs at the same instruction. Hence, this **does not** contradict the claim in 240-241: say, the agent has 10% of success rate for a particular instruction, then $V=0.1$. For successful rollouts, $A(s_h, a_h, c) = r(s_H, a_H, c) - V^\text{instruct}(c)=1-0.1=0.9$. But if the agent has 30% of success for an instruction, $A(s_h, a_h, c)=0.7$ for a successful rollout, because now $V=0.3$.
> ___
>
> ### How do you perform the train/test task split
>
> We use the standard train/test task split from AitW. Train and test tasks are generated following a set of same templates such as “Go to {shopping website}, and search for {item name}”, which might allow for generalization.

---

> ### Author Response · Authors · 2024-08-07
> **references**
>
> [1] ‘WebShop: Towards Scalable Real-World Web Interaction with Grounded Language Agents’. NeurIPS 2022.
>
> [2] Mind2Web: Towards a Generalist Agent for the Web’. NeurIPS 2023.
>
> [3] GPT-4V(Ision) Is a Generalist Web Agent, If Grounded. 2024, ICML 2024.
>
> [4] ‘TravelPlanner: A Benchmark for Real-World Planning with Language Agents’. ICML 2024.
>
> [5] Data-Driven Offline Optimization For Architecting Hardware Accelerators’. ICLR 2022.
>
> [6] Chip Placement with Deep Reinforcement Learning’. ICML 2022.
>
> [7] A Real-World WebAgent with Planning, Long Context Understanding, and Program Synthesis. ICLR 2024 (Oral).
>
> [8] Schulman, John, et al. ‘High-Dimensional Continuous Control Using Generalized Advantage Estimation’.

---

> ### Author Response · Authors · 2024-08-12
> **Discussion period ends soon**
>
> Dear reviewer 251x,
>
> Apologies for bothering you! Since the discussion period will end in two days, we would be grateful and would sincerely appreciate if you could respond to our rebuttal, leaving us enough time to address any remaining questions.
>
> Thanks, Authors

---

> ### Comment · Reviewer_251X · 2024-08-13
>
> Thank you for the rebuttal and I appreciate the efforts of providing many new experiments in the PDF, please do add these to the revised version. I think most of my concerns are properly addressed. Regarding the technical novelty, since my expertise mainly comes from RL algorithm research, and much less from developing AI agents/systems, I might not be in the best position to make the judgement.
>
> Nevertheless, given all information provided, I will raise my score 4->5 but lower my confidence 4->3, and vote for acceptance.

---

> > ### Author Response · Authors · 2024-08-13
> >
> > Thank the reviewer for reading our rebuttal We are glad that our rebuttal has solved most of your concerns. We appreciate your score raise and voting for acceptance!

---

### Official Review · Reviewer_XhdC · 2024-07-15

**Soundness:** 3
**Presentation:** 3
**Contribution:** 3
**Rating:** 6
**Confidence:** 3

**Summary:**

This paper proposes an autonomous RL approach, RL for digital agent (DigiRL), to finetune a pretrained VLM as an in-the-wild device control agent through GUI. The authors build a parallelizable Android learning environment with VLM-based evaluator to identify the key design choices for RL. The training include two stages an offline RL phase on existing data, then followed by an online RL phase by interacting with real-world graphical user interfaces using the Android learning environment. The proposed method with only 1.5B model size outperforms other state-of-the-art models such as GPT4-V or 17B CogAgent in the Android-in-the-Wild (AitW) dataset.

**Strengths:**

- The paper is well-structured and easy to follow.
- Many design choices are well motivated.
- The experiments are nice and well support the claims.

**Weaknesses:**

- Overall there is no major weakness. There are only several questions and potentially interesting empirical studies to look at. Check more details in the question sections.

**Questions:**

- Could the authors compare more advanced LLM reasoning or planning algorithms like Chain of Thoughts (CoT), Tree of Thoughts (ToT), Reasoning as Planning (RaP), etc.?
- Following the previous question, is it possible to compare with those planning/search-based methods with the autonomous evaluator or the trained value model of DigiRL as value functions?
- What is the training time and compute requirement for online training?
- In Figure 7, what does the AWR reweighting mean? Is it simply AWR?
- With the auto-curriculum setup, it may be interesting to look at what types of data/replay are critical throughout the online learning process; with simple categorization like failure mode in Figure 5 or whatever characterization can be interesting.
As detailed in the "Challenges of stochasticity" paragraph in section 3, could the authors provide some studies on unpredictable distractor and technical glitches?

**Limitations:**

Limitations are discussed in the paper.

---

> ### Author Rebuttal · Authors · 2024-08-07
>
> Thank you for your positive review and feedback on the paper. To address the raised questions, we add new results to include comparisons with  **CoT based planning**, and **a state-of-the-art LLM planning algorithm for device control called AppAgent [1]**. We also provide **additional results** for the auto-curriculum setup and ablation studies illustrating the challenge of stochasticity and dynamism for device control. We also provide clarifications on the rest of the questions, and will incorporate these results and clarifications in the final version of the paper.
>
> **Please let us know if your concerns are addressed and if so, we would appreciate it if you could upgrade your score. We are happy to discuss further.**  We answer your questions below:
>
> ___
>
> ### **[New result]** “Following the previous question, is it possible to compare with those planning/search-based methods with the autonomous evaluator or the trained value model of DigiRL as value functions?”
>
> Good question! In Table-3 of one-page PDF, we have included additional results of comparing planning/search-based methods with the autonomous evaluator. In particular, we compare with Reflexion[2] that for each task, first reflects on a trial run with the result given by the autonomous evaluator and performs a new trial with the reflection included in the prompt. We found that the use of Reflexion+autonomous evaluator can indeed enhance the performance (comparing GPT4V Reflexion+Set-Of-Marks 14.6% and GPT4V Set-Of-Marks 8.3%). However, **this approach still performs worse than our method DigiRL (14.6% compared to 67.2%)**.
>
> ___
>
> ### “Could the authors compare more advanced LLM reasoning or planning algorithms like Chain of Thoughts (CoT), Tree of Thoughts (ToT), Reasoning as Planning (RaP), etc.?”
>
> Thanks for bringing this up. We want to clarify that several baseline results in the submission were already prompted with a chain of thought. For instance, the “Set-of-Marks” approach with GPT4V and Gemini-1.5-Pro, and CogAgent are prompted with Chain of Thought (CoT) to produce an action. With regards to a SoTA planning baseline, we remark that we also compare DigiRL to AppAgent [1] in Table 1, which is a state-of-the-art VLM Retrieval Augmented Generation (RAG) approach specifically designed for device control. In both cases, we find that **DigiRL outperforms these prior methods**, indicating the superiority of training with autonomous RL for user-level device control over planning / prompting with frozen models.
>
> The methods discussed above constitute the state-of-the-art planning based approaches for device control. We are also happy to add any other comparisons if the reviewer has particular pointers for existing approaches that use planning / reasoning in the device-control domain.
>
> ___
>
> ### **[New result]** With the auto-curriculum setup, it may be interesting to look at what types of data/replay are critical throughout the online learning process; with simple categorization like failure mode in Figure 5 or whatever characterization can be interesting.
>
> We have run additional experiments to understand what types of data are being trained upon during the course of online learning. In particular, we categorize the tasks in the Web Shopping subset into three levels of difficulties according to the criterion in Table 3 in appendix, and plot the learning curve for each difficulty level during the online learning process in Figure 5 in Rebuttal PDF. From this plot, we can see that the performance of difficulty 1 (easy) tasks improves significantly in the first 200 trajectories indicating that much difficulty 1 data is trained upon in the very beginning. In the later stage of training, the performance of difficulty 1 tasks stay around the same while the performance of difficulty 2 and difficulty 3 tasks increase a lot, indicating data from difficulty 2 and difficulty 3 tasks is replayed more at this stage. This indicates that the auto-curriculum helps us upweight the right trajectories to focus at different stages of training.
>
> ___
>
> ### **[New result]** As detailed in the "Challenges of stochasticity" paragraph in section 3, could the authors provide some studies on unpredictable distractor and technical glitches?
>
> We have run an additional experiment where we take a trained DigiRL agent and continue to train it further with more online data for four days wall-clock time, even though the previous  training run that produced that agent had already converged. As shown in Figure 1 in the 1-page PDF, as more time passes the performance of the previously-trained DigiRL policy begins to drop. On the other hand, despite having converged in the previous training run, continuing to update the DigiRL agent with more online interactions lead to more robust and stable performance. We believe that this experiment precisely illustrates the issues with stochasticity and dynamic nature of websites: the decay in performance of the optimal policy trained previously underscores the challenge of dynamism and stochasticity of the dynamics of the device control setup, and stable performance with continued DigiRL training demonstrates the efficacy of our approach in enabling performance despite these challenges.
>
> ___
>
> ### In Figure 7, what does the AWR reweighting mean? Is it simply AWR?
>
> Yes you are correct it is simply AWR. We call it reweighting as it uses reweights to “soft filter” advantageous actions while we use a “hard filtering” for implementation simplicity.
>
> ___
>
> ### What is the training time and compute requirement for online training?
>
> For our main experiments, we are able to run 8 emulators in parallel on a 16GB T4 GPU with 32 CPUs, and finish an online training run with 1k trajectories within 3 days. Figure 16 in appendix illustrates a relative speed up obtained if we use more resources. For example, if we use 128 CPUs and 4 GPU, we can achieve a speed up of 3.55x when it is set up properly.

---

> ### Author Response · Authors · 2024-08-07
> **references**
>
> [1] Zhang, Chi, et al. ‘AppAgent: Multimodal Agents as Smartphone Users’.
>
> [2] Shinn, Noah, et al. ‘Reflexion: Language Agents with Verbal Reinforcement Learning’.

---

> > ### Comment · Reviewer_XhdC · 2024-08-13
> > **Thanks for the rebuttal**
> >
> > The rebuttal addressed most of my previous concern. Thanks!

---

> > > ### Author Response · Authors · 2024-08-13
> > >
> > > We thank the reviewer for recognizing our efforts in the rebuttal and additional new results to address your previous concerns, and are glad that the additional results address the concerns. Since there is still one more day, we are also wondering if there would be some other discussion or evidence that we can provide in this period to help improve your score of our paper further. Please let us know. We would be very grateful if you are willing to upgrade your score. Thanks a lot!

---

### Official Review · Reviewer_JuUV · 2024-07-17

**Soundness:** 3
**Presentation:** 3
**Contribution:** 2
**Rating:** 6
**Confidence:** 3

**Summary:**

This paper tackles AI agent training for controlling digital devices (e.g., web navigation). The proposed framework, named DigiRL, is a 3-stage training process consisting of model pre-training, offline fine-tuning (offline RL), and online fine-tuning (online RL). To achieve this goal, the authors first build a parallelizable Android learning environment that enables fast online interactions for policy learning; they then adopt a VLM-based evaluator to provide reward signals for the agents; finally, they perform ablation studies to examine several key design choices in typical policy-based RL methods for the third stage. Compared to larger models trained without this stage, the proposed approach enjoys significant performance enhancement due to the online fine-tuning stage.

**Strengths:**

+ The authors did a good job introducing the background, the problem setup, the baselines, and the details of their proposed method.
+ Fine-tuning large VLMs in an online fashion can be challenging; the performance improvement obtained by the proposed method, which is relatively simple, is substantial and the overall approach looks promising.

**Weaknesses:**

The main issue is the limited comparison with online RL methods for fine-tuning LLM-based agents. The only RL method compared in the experiments is Filtered BC (besides vanilla AWR, which the proposed method is based on). Filtered BC is strictly speaking not an online RL method. Admittedly, AI agent training for device control is a relatively under-explored new area, and the authors claim that theirs is the first successful offline-to-online RL approach for device control AI agents, I believe more experiments with other online RL baselines not originally designed for device control is required to justify DigiRL's advantages. For example, the classic on-policy methods such as REINFORCE and PPO, or the more recent ones that are more sample-efficient, such as [1, 2] (which might be considered more-or-less recurrent work, though). Further comparisons also help to provide more insight into the unique challenges of the device control problem for digital agents.

[1] Trial and Error: Exploration-Based Trajectory Optimization for LLM Agents
[2] REBEL: Reinforcement Learning via Regressing Relative Rewards

**Questions:**

- Will the proposed method further scale well with more online interactions?
- While multi-turn interactions are a challenge of the device control problem, is there any component in the proposed framework that specifically helps to tackle it?

I am willing to adjust my ratings after seeing the authors' responses.

**Limitations:**

Yes.

---

> ### Author Rebuttal · Authors · 2024-08-07
>
> Thank you for your review and feedback on our paper. To address the main concern regarding comparisons, we provide additional results and appeal to comparisons from prior work to demonstrate that DigiRL outperforms several online RL methods. These comparisons include REINFORCE, PPO, and Q-value based methods. We also provide **additional results** to show the favorable scaling of our method with more interactions. **Please let us know if your concerns are addressed and if so, we would appreciate it if you might be willing to upgrade your score. We are happy to discuss further.**  We answer your questions below:
>
> ___
>
> ### **[New result] Comparisons to other online RL methods**
>
> We also provide new results for a comparison against several online RL methods.
>
> 1) **REINFORCE**: We note that policy gradient via REINFORCE reduces to an online filtered BC loss when the rewards are given by +1 and 0, as a result, our filtered BC results in the paper already indicate that DigiRL outperforms REINFORCE. To see this equivalence between REINFORCE and online filtered BC, note that:
> $ L_\text{REINFORCE} = E_{\tau \sim \mathcal{D}}(\sum_{h=1}^{H}r_h)(\sum_{h=1}^{H}\log \pi(a_h|s_h))$ (surrogate for REINFORCE)
> and
> $L_\text{Filtered-BC} =E_{\tau \sim \mathcal{D}}\mathbb{1}\{\sum_{h=1}^{H}r_h > \text{t}\}(\sum_{h=1}^{H}\log \pi(a_h|s_h))$, for trajectories $\tau$ and threshold $t$.
>
> Since our reward takes a binary 0/1 value at the end of a rollout; both $(\sum_{h=1}^{H}r_h)$ and $\mathbb{1}\{\sum_{h=1}^{H}r_h > \text{thresh}\}$ will only evaluate to 1 if the trajectory is successful or be 0 otherwise, resulting in an identical multiplication factor for both REINFORCE and online filtered BC. Our results already show that DigiRL outperforms online filtered BC, which should imply that DigiRL outperforms REINFORCE as well.
>
> 2) **PPO**: We have run experiments comparing DigiRL with online PPO, and we found that DigiRL is more efficient than PPO. The efficiency of DigiRL compared PPO stems from 1) DigiRL starts from an offline RL checkpoint so that it maintains an initial advantage over PPO via the use of offline data, and 2) DigiRL is able to make use of off-policy data to stabilize training while PPO always updates on the small batch of on-policy data. The inefficiency of on-policy PPO corroborates findings in recent multi-turn RL literature [1,2]. We will run this comparison fully and add this baseline in all settings.
>
> 3) **Q-value based actor-critic methods.** We have also tried some other designs involving Q-function training [1, 2] for the step-level advantage estimator and found that training a Q-function in all of our experiments obtained inferior performance. Concretely, we attempted to use $Q(s, a) - V(s)$ to compute step-level advantages following Zhou et al. [2]. In this case, we were not able to get the Q-function to correctly pay attention to the action input, leading to Q and V collapsing to very similar values everywhere. We hypothesize this is because it is relatively easy to understand what elements are present in a screenshot, but learning how an action, which appears as a small change on the screen affects the screenshot is more challenging because this requires inferring relationships between precise relative locations of each element and clicking coordinates.
>
> As shown in Table 1 of the 1-page PDF for the Web Shopping subset, the design choice of using $V(s’) + r - V(s)$ to calculate one-step advantage to instead of $Q(s,a) - V(s)$ led to better offline RL performance by 9% and reduced variance by 5.9%. Of course, this does not mean that Q-functions cannot be trained on this task, but that within the time of the rebuttal period, we found it quite hard to get reasonable Q-functions needed for value-based online RL methods.
>
> Finally, thanks for bringing up the concurrent work that we will discuss. We are happy to explore such methods in an updated version of the paper.
> ___
>
> ### **[New result] Scaling with more online interactions?**
>
> In Figure 8 of the submission, we provide a learning curve plotting performance as more online data is collected. We also include a new result Figure 1 of 1-page PDF, where we set the agent to be updated with even more online data for four days after convergence. Compared to a frozen policy, the agent trained with more online interactions data maintains a stable performance despite the changing nature of the websites and the device state, while the performance of a frozen RL policy gradually decays as time goes on. **This indicates that DigiRL utilizes online interaction effectively.**
>
> ___
>
> ### “While multi-turn interactions are a challenge of the device control problem, is there any component in the proposed framework that specifically helps to tackle it?”
>
> The design of the doubly robust estimator for estimating step-level advantage that balances bias and variance is specifically useful in multi-turn settings, in stochastic environments. Such a design for variance reduction has been shown to be unnecessary in the single-turn setting [3], but our prior results on comparing a one-step advantage estimator $V(s’) + r - V(s)$ and doubly-robust advantage estimator (Eqn. 4.3 in paper) show that it is critical.
>
> Likewise, the use of a curriculum is especially important in highly multi-task environments, where training uniformly on all initial states is likely to not provide a strong signal to update the policy [4] (as shown in the comparison between “Ours w/ step-level advantage” and “Filtered BC” in Fig 7 in paper). We will clarify this.
>
>
> [1] Song, Yifan, et al. ‘Trial and Error: Exploration-Based Trajectory Optimization for LLM Agents’. ACL 2024
>
> [2] Zhou, Yifei, et al. ‘ArCHer: Training Language Model Agents via Hierarchical Multi-Turn RL’. ICML 2024
>
> [3] Ahmadian, Arash, et al. ‘Back to Basics: Revisiting REINFORCE Style Optimization for Learning from Human Feedback in LLMs’.
>
> [4] Jiang, Minqi, et al. ‘Prioritized Level Replay’.

---

> ### Comment · Reviewer_JuUV · 2024-08-14
> **Raised my score**
>
> Thanks for the rebuttal. I have carefully read all the responses and reviews. I will raise my score. With that being said, I will leave it up to the AC to decide whether the contribution of developing agents is strong enough for the paper's acceptance. I also lowered my confidence as my expertise mostly lies in RL.

---

### Author Rebuttal · Authors · 2024-08-07

We would like to thank all the reviewers for their feedback. We are glad that the Reviewer Xhdc thinks that there is  “no major weakness” in the paper and that Reviewer zDzF thinks that “the usage of RL for designing a successful agent for device control tasks is fascinating”.

At the outset, we would like to clarify that our goal is to show that autonomous RL can be used to build SOTA device control agents that outperform the dominant approaches in the device-control community including prompting proprietary models (Gemini/GPT-4) and fine-tuning with human demonstrations. Rather than claiming novelty of the algorithm, the main contribution of this work is the efficacy of our effective RL system / approach including the environment setup, usage of autonomous evaluator, and particular design choices specific to the real-world challenges of this environment. On the RL side, our methodological contribution involves identifying and designing the right RL objective to be able to do so (robust advantages + curriculum + cross-entropy loss + AWR). (see Fig.7 in paper and one-page PDF).

In the rebuttal period, we have run additional experiments and plot the results in the one-page pdf, including:
- Table 1: Ablation on Q-function based advantage estimations (Reviewer JuUv, Xhdc, and 251x)
- Table 2: Hyperparameter profiling for step-level advantage estimation (Reviewer 251x)
- Table 3: Comparison with search-based method with autonomous evaluator (Reviewer Xhdc)
- Figure 1: Study on the effect of stochasticity (Reviewer JuUv and 251x)
- Figure 2: Algorithm diagram (Reviewer zDzF)
- Figure 3: Comparison with pure-online setting (Reviewer 251x)
- Figure 4: Comparison with pure-online PPO (Reviewer JuUv)
- Figure 5: Studies on auto-curriculum (Reviewer Xhdc)

We thank the reviewers in advance for paying attention to our new results and clarifications. We look forward to discussions and hope that our responses and the discussion will convince the reviewers that our work is valuable.

---

> ### Author Response · Authors · 2024-08-11
>
> Dear reviewers,
>
> Apologies for bothering you! Since we are getting close to the end of the discussion period, we would be grateful and would sincerely appreciate if you could respond to our rebuttal, leaving us enough time to address any remaining questions.
>
> Thanks, Authors

---

### Decision · Program_Chairs · 2024-09-25

**Decision:**

Accept (poster)

**Comment:**

Thank you for the rebuttal. After reviewing the reviewers' comments and the authors' responses, I acknowledge the contributions of this paper in advancing the use of Reinforcement Learning (RL) for digital agent control through Graphical User Interfaces (GUIs). The introduction of DigiRL, a novel autonomous RL approach combining offline training on static demonstrations with an in-the-wild, offline-to-online RL stage, offers a compelling solution to real-world stochasticity in device control tasks. I appreciate the authors addressing the major concerns during the rebuttal and recommend accepting this paper.